# Sequential Multi-Agent Dynamic Algorithm Configuration

**Chen Lu**[1,2], **Ke Xue**[1,2]*, **Lei Yuan**[1,2], **Yao Wang**[3],
**Yaoyuan Wang**[3], **Sheng Fu**[3], **Chao Qian**[1,2]*

[1] National Key Laboratory for Novel Software Technology, Nanjing University, China
[2] School of Artificial Intelligence, Nanjing University, China
[3] Advanced Computing and Storage Lab, Huawei Technologies Co., Ltd. Shenzhen, China
`{xuek, qianc}@lamda.nju.edu.cn`

## Abstract

The performance of an algorithm often critically depends on its hyperparameter configuration. Dynamic algorithm configuration (DAC) is a recent trend in automated machine learning, which can dynamically adjust the algorithm's configuration during the execution process and relieve users from tedious trial-and-error tuning tasks. Recently, multi-agent reinforcement learning (MARL) approaches have improved the configuration of multiple heterogeneous hyperparameters, making various parameter configurations for complex algorithms possible. However, many complex algorithms have inherent inter-dependencies among multiple parameters (e.g., determining the operator type first and then the operator's parameter), which are, however, not considered in previous approaches, thus leading to sub-optimal results. In this paper, we propose the sequential multi-agent DAC (Seq-MADAC) framework to address this issue by considering the inherent inter-dependencies of multiple parameters. Specifically, we propose a sequential advantage decomposition network, which can leverage action-order information through sequential advantage decomposition. Experiments from synthetic functions to the configuration of multi-objective optimization algorithms demonstrate Seq-MADAC's superior performance over state-of-the-art MARL methods and show strong generalization across problem classes. Seq-MADAC establishes a new paradigm for the widespread dependency-aware automated algorithm configuration. Our code is available at `https://github.com/lamda-bbo/seq-madac`.

## 1 Introduction

Identifying proper configurations of hyperparameters is critical for many learning and optimization algorithms [15, 26]. Algorithm configuration (AC) [14, 23] has emerged to alleviate the user's burden of manual trial-and-error tuning. However, the static configuration policies obtained by AC may not achieve optimal performance, because algorithms may require different configurations at different execution stages [37]. Dynamic AC (DAC) [2, 1] is a prevalent paradigm that enables dynamic adaptation of the algorithm's configuration during the execution process, which is more flexible compared to AC. Specifically, DAC formulates the configuration process as a contextual Markov decision process (MDP) and then leverages reinforcement learning (RL) [32] to learn dynamic configuration policies. DAC has been shown to outperform static AC methods on many tasks, such as learning rate tuning of deep neural networks [7] and step-size control of evolution strategies [35].

Due to the increasing complexity of real-world problem modeling, the performance of an algorithm usually relies on multiple types of hyperparameters. Consider MOEA/D [48], a widely adopted

---

*Corresponding Author

evolutionary algorithm [49] for multi-objective optimization problems, as a case in point. It involves four categories of configuration hyperparameters: weights, neighborhood size, reproduction operator type, and parameters associated with the utilized reproduction operator. Each of these hyperparameters is critical and exerts a significant impact on the algorithm's performance [41]. For this challenging task of tuning multiple hyperparameters, traditional DAC methods tend to be ineffective [35, 42]. These methods typically tune one hyperparameter type while freezing the rest, a limitation that prevents them from capturing the inter-relationships and dependencies across different hyperparameter types. Recently, cooperative multi-agent RL (MARL) [47] approaches have been proposed to enhance the dynamic configuration problem of multiple heterogeneous hyperparameters by modeling it as a contextual multi-agent MDP (MMDP) [44], broadening DAC's application scope and enabling various hyperparameter configurations for complex algorithms.

In addition to multiple complex hyperparameters, another important barrier of real-world DAC problems is that these hyperparameters are not completely decoupled, which may have inherent inter-dependencies. For instance, when configuring MOEA/D for multi-objective optimization, once the reproduction operator type is selected, the parameters associated with the utilized reproduction operator can subsequently be determined. Unfortunately, previous approaches do not consider this inherent property. Besides, current advanced MARL algorithms mainly focus on issues such as coordination and adaptation mechanisms in non-stationary environment [47, 31, 46, 17, 22], which are not specifically designed to handle the inherent inter-dependencies among agents.

In this paper, we propose a sequential multi-agent DAC (Seq-MADAC) framework to address this issue by considering the inherent inter-dependencies of multiple parameters. Specifically, we formulate this task as a contextual sequential MMDP and propose a sequential advantage decomposition network (SADN), which can leverage action-order information through sequential advantage decomposition. We also theoretically demonstrate the rationality of SADN based on the individual global max (IGM) principle [36]. Leveraging this beneficial property, SADN achieves effective sequential decomposition of the demanding multiple-hyperparameter DAC task for joint action optimization, significantly easing the issues of combinatorial explosion of the action space and complex inter-dependencies. Empirically, we compare SADN with other methods that consider sequential actions, including ACE [19] and SAQL [3], as well as a range of general advanced MARL algorithms, including VDN [39], QMIX [31], MAPPO [46], HAPPO [17], and HASAC [22]. On controllable synthetic functions, we demonstrate that in higher-dimensional problems and noisy scenarios, SADN shows a more pronounced advantage in optimization efficiency and robustness. On more complex and challenging multi-objective optimization problems, SADN shows superior performance over state-of-the-art MARL methods and demonstrates strong generalization across problem classes.

Our contributions include three perspectives:

1. Formulating a sequential decision-making framework to model the DAC task with multiple hyperparameters and inter-dependencies as a contextual sequential MMDP, enabling sequential action ordering in dynamic hyperparameter configuration.

2. Introducing SADN with sequential advantage decomposition to exploit action-order information for improved coordination and better credit assignment.

3. Conducting extensive experiments on both simple synthetic and complex multi-objective optimization problems to validate SADN's effectiveness in superior optimization performance and robust generalization capability.

## 2 Background

### 2.1 Dynamic Algorithm Configuration

Compared to the static configuration scheme of AC, DAC focuses on dynamically adjusting algorithm configuration throughout the optimization process, which can be formulated as a contextual MDP $\mathcal{M}_{\mathcal{I}} := \mathcal{M}_{i \sim \mathcal{I}}$ [2], and can be addressed by leveraging RL techniques. Here, $\mathcal{I}$ represents the problem instance space, and each $\mathcal{M}_i := \langle \mathcal{S}, \mathcal{A}, \mathcal{T}_i, r_i \rangle$ corresponds to one target problem instance $i \in \mathcal{I}$ [2, 9]. Such a context notion $\mathcal{I}$ enables studying policy generalization [16]. For a target algorithm $A$ with configuration hyperparameter space $\Theta$, a DAC policy $\pi \in \Pi$ takes the state $s \in \mathcal{S}$ as input and outputs an action $a \in \mathcal{A}$. Here, the state $s \in \mathcal{S}$ typically represents the historical performance changes of algorithm $A$, while the action $a \in \mathcal{A}$ corresponds to a hyperparameter

configuration $\theta \in \Theta$ of algorithm $A$. DAC aims at improving the performance of algorithm $A$ on a set of instances (e.g., optimization functions). Given a probability distribution $p$ over the instance space $\mathcal{I}$, the objective of DAC is to identify an optimal policy $\pi^*$:

$$\pi^* \in \arg\min_{\pi \in \Pi} \int_{i \in \mathcal{I}} p(i)c(\pi, i)\mathrm{d}i,$$

where $i \in \mathcal{I}$ is an instance to be optimized, and $c(\pi, i) \in \mathbb{R}$ is the cost function of the target algorithm with policy $\pi$ on the instance $i$. Previous works have shown the superiority of DAC compared to the static policies in many scenarios, e.g., learning rate adaptation in SGD [7], step-size adaptation in CMA-ES [35], and heuristic selection in planning [37]. However, both traditional DAC methods and these applications only involve a single type of hyperparameter. Dynamic configurations of complex algorithms with multiple types of hyperparameters have been found to be difficult [9, 1].

MADAC [44] aims to simultaneously configure multiple hyperparameters of different types, in order to cope with the increasing complexity of algorithm structure and the increasing number of hyperparameters. To address this issue, MADAC models it as a cooperative multi-agent problem with one agent configuring one parameter, to take the interactions of different parameters into account. Another advantage of the cooperative multi-agent modeling framework is its capacity to address the combinatorial explosion challenge associated with joint action spaces. However, in practice, parameters may have inherent inter-dependencies on each other (e.g., determining the operator type first and then the operator parameters [12]), which is not considered in MADAC. It is quite important to take these inherent inter-dependencies into consideration, since we can explore the configuration space more effectively without exploring illegal combinations, which may bring better results.

Recently, coupled action-dimensions with importance differences (CANDID) DAC [3] considers the interaction effects among hyperparameters exhibiting different levels of importance, assuming that the important parameters should be executed first. However, in practice, parameters exhibit complex inherent inter-dependencies. Consequently, important parameters should not be prioritized for adjustment if they depend on prior parameters. For example, while the operator parameter holds greater importance, optimized performance is attained by first confirming the operator type before adjusting the operator parameter. Moreover, CANDID DAC tries to address this issue by extending independent Q-learning (IQL) [40] to sequential agent Q-learning (SAQL), i.e., adding the actions made by prior agents to the state of subsequent agents. However, this may suffer from similar disadvantages to IQL: The team reward is used directly in the individual agent's training and does not consider the credit assignment among agents, thus it cannot reveal the interactions between the agents. What's worse, it may lead to the non-convergence issue [31], because each agent's learning is interfered with by the learning process of others, and the team reward cannot provide the corresponding guidance. Thus, how to propose a framework that can efficiently leverage the internal relationships of the problem, including inherent parameter inter-dependencies and interactions among agents, remains an ongoing challenge.

## 2.2 Multi-Agent Reinforcement Learning

A fully observable cooperative multi-agent system [45] can be formulated as an MMDP [5], defined as $\mathcal{M} := \langle \mathcal{N}, \mathcal{S}, \{\mathcal{A}_j\}_{j=1}^n, \mathcal{T}, r \rangle$, where $\mathcal{N}$ represents $n$ agents, $\mathcal{S}$ is the state space, and $\mathcal{A}_j$ is agent $j$'s action space. At each time-step, agent $j \in \mathcal{N}$ acquires $s \in \mathcal{S}$ and then chooses an action $a^{(j)} \in \mathcal{A}_j$. The joint action $\boldsymbol{a} = \langle a^{(1)}, \ldots, a^{(n)} \rangle$ leads to next state $s' \sim \mathcal{T}(\cdot \mid s, \boldsymbol{a})$ with a shared reward $r(s, \boldsymbol{a})$. The goal of an MMDP is to find a joint policy that maps the states to probability distributions over joint actions, $\boldsymbol{\pi} : S \to \Delta(\mathcal{A}_1 \times \mathcal{A}_2 \times \cdots \times \mathcal{A}_n)$, where $\Delta(\mathcal{A}_1 \times \mathcal{A}_2 \times \cdots \times \mathcal{A}_n)$ stands for the distribution over joint actions, with the goal of maximizing the global value function: $Q^{\boldsymbol{\pi}}(s, \boldsymbol{a}) = \mathbb{E}_{\boldsymbol{\pi}} \left[ \sum_{t=0}^{\infty} \gamma^t r(s_t, \boldsymbol{a}_t) \mid s_0 = s, \boldsymbol{a}_0 = \boldsymbol{a} \right].$

Many algorithms have been proposed to solve the cooperative MARL [47, 10], which can be basically divided into two categories: value-based methods and policy gradient-based methods. Among the value-based methods, VDN [39] addresses the problem of multi-agent collaboration through value function decomposition. It makes a main assumption that the team Q-value function can be additively decomposed into the simple sum of the local Q-values of each agent: $Q(s, \boldsymbol{a}) \approx \sum_{i=1}^n Q_i(s, a_i)$, where $Q(s, \boldsymbol{a})$ is the joint action value function for the whole team, $s$ is the shared state, $\boldsymbol{a}$ is the joint action, $n$ is the number of agents, $Q_i(s, a_i)$ is the agent $i$'s local individual action value function, and $a_i$ is the action taken by agent $i$. One critical principle the value decomposition methods need to satisfy is the Individual Global Max (IGM) principle defined in Definition 1.

**Definition 1** (Individual Global Max [36]). *For a joint action-value function $Q : S \times \mathcal{A} \rightarrow \mathbb{R}$, if there exist individual action-value functions $[Q_i : S \times \mathcal{A}_i \rightarrow \mathbb{R}]_{i=1}^n$, such that the following holds:*

$$\arg\max_{\boldsymbol{a}} Q(s, \boldsymbol{a}) = \begin{pmatrix} \arg\max_{a_1} Q_1(s, a_1) \\ \arg\max_{a_2} Q_2(s, a_2) \\ \vdots \\ \arg\max_{a_n} Q_n(s, a_n) \end{pmatrix}$$

*then, we say that $[Q_i]$ satisfy IGM for Q, which means the decomposition satisfies the IGM principle.*

This critical principle guarantees the optimal efficient decentralized execution, as each agent only needs to choose its own optimal action regardless of other agents' actions. Obviously, VDN's additive decomposition satisfies the IGM principle, but is restricted to its additive form, failing to model the complex situations in many cooperative MARL scenarios. QMIX [31] uses a mixing network with non-negative weights to learn a monotonic combination of local individual action value functions as the joint action value function for the whole team, which extends the forms of factorization.

Another type of cooperative MARL algorithm is the policy gradient-based method. MAPPO [46] extends the popular single-agent RL algorithm proximal policy optimization (PPO) [34] to the multi-agent RL setting and achieves surprising effectiveness. However, MAPPO is mainly designed for homogeneous agents with shared action space and policy parameters. HAPPO [17] further addresses heterogeneous agents with a theoretical property of monotonic improvement guarantee. HASAC [22] obtains the maximum entropy objective for MARL from probabilistic graphical models to escape from converging to a suboptimal Nash Equilibrium. A2PO [30] is an agent-by-agent sequential update algorithm with a theoretical guarantee of the monotonic policy improvement that accelerates optimization with a semi-greedy agent selection rule. However, these recent works do not fit the needs of DAC well. Their execution is fully decentralized, with each agent unaware of others' current actions, which ignores the inherent inter-dependencies between the agents' actions.

ACE [19] models the multi-agent decision-making problem into a sequential decision-making problem with the agents taking action one by one, formulating this as a sequentially expanded MDP (SE-MDP). At each timestep, the agents make decisions based on the actions taken by their prior agents at the current timestep, and the action value function of each individual agent is updated according to the performance of their subsequent agents, which make decisions based on the taken action. The update scheme of action value functions is as follows:

$$Q_i(s, \boldsymbol{a}_{1:i-1}, a_i) \leftarrow \begin{cases} \max_{a_{i+1}} Q_{i+1}(s, \boldsymbol{a}_{1:i}, a_{i+1}), & \text{if } i < n \\ r + \gamma \max_{a_1'} Q_1(s', a_1') & \text{if } i = n \end{cases}$$

The sequential modeling relieves the non-stationary training problem in MARL because the agents are aware of their prior agents' actions, which also fits the need of DAC for being able to capture the inherent inter-dependencies between the agents' actions. However, the update scheme of ACE can suffer from fragile training due to its long sequence of action value function update. That is, if some agents fail or fall into local optima within the long action value function update chain, the learning of the whole chain will be affected and even damaged.

Recently, similar ideas of sequential action choosing have been applied to offline RL in the centralized setting. Q-Transformer [6] and Q-Mamba [24] have been proposed for effective offline RL training in a centralized manner, where advanced neural network architectures like transformer and Mamba are integrated with the sequential Q function update scheme similar to ACE.

## 3 Method

### 3.1 Contextual Sequential Multi-Agent MDP

Previous works have paid attention to sequential modeling to cope with the high-dimensional action spaces (Sequential MDP [27]) and non-stationary training (SE-MDP [19]) in MARL. The main idea is to break the global state transition into a sequence of intermediate decision-making steps and select the current timestep's action one dimension by one dimension. That is, the original transition $(s, \boldsymbol{a}, s')$ is transformed into $n$ intermediate decision making steps $(s, a_1), (s, a_1, a_2), \dots, (s, a_1, a_2, \dots, a_n, s')$, where $n$ is the number of action dimension, and when selecting the $i$-th action $a_i$ for dimension

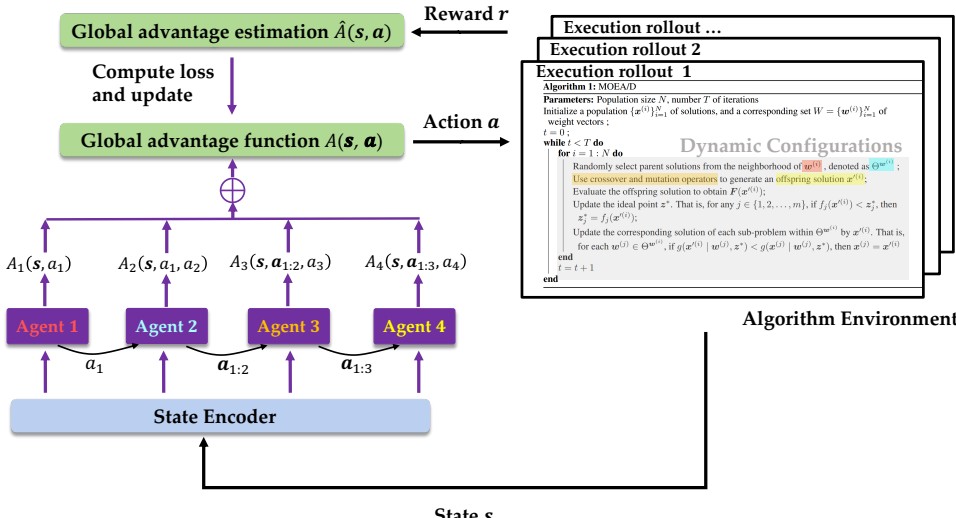

Figure 1: Workflow of the proposed sequential advantage decomposition network (SADN).

$i$, the previous selected actions $\boldsymbol{a}_{1:i-1}$ for dimension 1 to $i-1$ are available for decision making. Prior work shows that under the CANDID properties [3], sequential policies can coordinate action selection between dimensions while avoiding the combinatorial explosion of the action space. Their results encourage an extended study of sequential policies for DAC.

However, the previous sequential modeling is typically a single-agent MDP, with one agent selecting actions one dimension by one dimension, which fails to model the complex cooperation and communication among heterogeneous agents. Moreover, the different contributions of different action dimensions to the global reward are neglected in the sequential single-agent MDP modeling, which harms the learning efficiency and may lead to sub-optimal results.

In the DAC task, there are often different heterogeneous parameters (i.e., they are of different types and have different effects on the final performance of the target algorithm) with complex inherent inter-dependencies, which call for cooperation and communication among the tuning of different parameters. What's more, as the heterogeneous parameters have different effects on the target algorithm's performance, the tuning of them should be guided with differentiated credit assignment according to their contribution to the improvement of the target algorithm's performance.

Therefore, we formulate the Seq-MADAC task as a contextual sequential MMDP as $\mathcal{M}_{\mathcal{I}} := \mathcal{M}_{i \sim \mathcal{I}}$, where $\mathcal{I}$ represents the space of problem instances, and $\mathcal{M}_i := \langle \mathcal{N}, \mathcal{S}, \{\mathcal{A}_j\}_{j=1}^n, \mathcal{T}_i, r_i \rangle$ is each sequential MMDP corresponding to one target problem instance $i \in \mathcal{I}$. In a sequential MMDP, at the timestep $t$, the agents are allowed to sequentially take their actions and communicate with each other to make better decisions. That is, the available information for the $i$-th agent at timestep $t$ is $(s_t, a_1^t, a_2^t, \ldots, a_{i-1}^t) = (s_t, \boldsymbol{a}_{1:i-1}^t)$, and the $i$-th agent selects its action $a_i^t$ accordingly. When all the agents finish selecting their actions, the state transforms to the next state $s_{t+1} \sim \mathcal{T}(\cdot \mid s_t, \boldsymbol{a}^t)$ based on the joint action $\boldsymbol{a}^t = (a_1^t, \ldots, a_n^t) = \boldsymbol{a}_{1:n}^t$, and all agents get a shared global reward $r(s_t, \boldsymbol{a}^t)$.

## 3.2 Sequential Advantage Decomposition Network

Recent sequential update schemes in MARL suffer from interference from other agents' learning (i.e., SAQL [3]) or the fragility of the long action value function update chain (i.e., ACE [19]), while the Seq-MADAC task requires MMDP modeling with differentiated credit assignment. Thus, we propose a decomposition-based method called Sequential Advantage Decomposition Network (SADN) to address these issues, whose decomposition does not need any assumption, with the IGM principle satisfied. The workflow of our SADN is shown in Figure 1.

Firstly, we give the definition of the action value function and the advantage function of each agent in the sequential setting. We define the action value function for the previous $k$ agents at state $s$ as $Q_{1:k}^{\boldsymbol{\pi}}(s, \boldsymbol{a}_{1:k})$, and the $k$-th agent's advantage function based on state $s$ and the actions $\boldsymbol{a}_{1:k-1}$ taken by its prior agents as $A_k^{\boldsymbol{\pi}}(s, \boldsymbol{a}_{1:k-1}, a_k)$ in the sequential form in Definition 2 following [17].

**Definition 2** ([17]). *Let $n$ be the total number of agents, $\boldsymbol{a}_{1:k}$ denote the actions taken by agent 1 to $k$ at the current timestep, and $Q^{\boldsymbol{\pi}}(s, \boldsymbol{a})$ denote the global action value function. The corresponding multi-agent state-action value function is defined as*

$$Q_{1:k}^{\boldsymbol{\pi}}(s, \boldsymbol{a}_{1:k}) := \mathbb{E}_{\boldsymbol{a}_{k+1:n} \sim \boldsymbol{\pi}_{k+1:n}}[Q^{\boldsymbol{\pi}}(s, \boldsymbol{a}_{1:k}, \boldsymbol{a}_{k+1:n})],$$

*and the advantage function of agent $k$ with the given state $s$ and the actions $\boldsymbol{a}_{1:k-1}$ taken by its prior agents is*

$$A_k^{\boldsymbol{\pi}}(s, \boldsymbol{a}_{1:k-1}, a_k) := Q_{1:k}^{\boldsymbol{\pi}}(s, \boldsymbol{a}_{1:k}) - Q_{1:k-1}^{\boldsymbol{\pi}}(s, \boldsymbol{a}_{1:k-1}).$$

Following Definition 2, we give the multi-agent advantage decomposition lemma [17] in the sequential form in Lemma 1. For the proof, please see Appendix A.3.

**Lemma 1** (Multi-Agent Advantage Decomposition [17]). *In any cooperative Markov game, given a joint policy $\boldsymbol{\pi}$, the global advantage function $A^{\boldsymbol{\pi}}(s, \boldsymbol{a})$, and $n$ agents in total, for any state $s$, the following equations hold:*

$$A^{\boldsymbol{\pi}}(s, \boldsymbol{a}) = \sum_{i=1}^{n} A_i^{\boldsymbol{\pi}}(s, \boldsymbol{a}_{1:i-1}, a_i).$$

Inspired by the action value function decomposition and the implicit reward assignment by using a decomposition network proposed in [39], as well as the multi-agent advantage decomposition lemma in [17], we decompose the advantage function sequentially. In specific, the $i$-th agent maintains a network that models the $i$-th advantage function $A^{\boldsymbol{\pi}_i}(s, \boldsymbol{a}_{1:i-1}, a_i)$ and selects the action that maximizes it, and we compute the global advantage function by the sum of all individual advantage functions, using Lemma 1. The $i$-th advantage function is updated implicitly when the global advantage function is updated by backpropagating gradients. To update the global advantage function, we maintain a global value network, like the critic in the actor-critic framework, and compute the target global advantage function by Generalized Advantage Estimation (GAE, [33]) with $\lambda = 0$ (i.e., one-step temporal difference), which corresponds to the update scheme of action value function in Q-learning (the detailed proof see Appendix A.1). The global advantage function update scheme is as follows:

$$A(s, \boldsymbol{a}) \leftarrow A(s, \boldsymbol{a}) + \alpha \cdot [r + \gamma V(s') - V(s) - A(s, \boldsymbol{a})],$$

where $A(s, \boldsymbol{a})$ is the global advantage function, $s$ is the current state, $\boldsymbol{a}$ is the joint action, $s'$ is the next state, $\alpha$ is the steplength, $\gamma$ is the discount factor, and $V(s)$ is the global value function.

Our method also satisfies the IGM principle in the sequential setting, which is stated in Theorem 1.

**Theorem 1.** *Selecting the best joint action $\boldsymbol{a}$ for state $s$ to maximize the global action value function is equivalent to sequentially selecting the best action $a_i$ for state $s$ to maximize the agent $i$'s advantage function. That is,*

$$\arg\max_{\boldsymbol{a}} Q(s, \boldsymbol{a}) = \begin{pmatrix} \arg\max_{a_1} A_1(s, a_1) \\ \arg\max_{a_2} A_2(s, \arg\max_{a_1} A_1(s, a_1), a_2) \\ \vdots \\ \arg\max_{a_n} A_n(s, \arg\max_{a_1} A_1(s, a_1), \arg\max_{a_2} A_2(s, \arg\max_{a_1} A_1(s, a_1), a_2), \dots, a_n) \end{pmatrix}.$$

With this beneficial property, SADN provides an efficient and effective sequential decomposition of the challenging high-dimensional task of joint action optimization, where the individual agent only needs to focus on optimizing its own individual advantage function by taking its own action. It is of great benefit because the IGM principle guarantees the consistency between the joint action optimality and the individual action optimality, which significantly alleviates the problem of combinatorial explosion of the action space and complex inter-dependencies. Detailed proofs are provided in Appendix A.2 due to space limitation.

The advantage decomposition in the SADN allows each individual advantage net to be updated independently and simultaneously, reducing their mutual influence and compounding errors. In

contrast, ACE sequentially updates the Q nets based on the output of the previous agent (i.e., more centralized), which leads to compounding errors along the update chain, especially worsening in the long sequence updates. Moreover, SADN decomposes the global advantage function sequentially, where the individual advantages are learned relatively independently, and the action sequence information is also captured in the update order of the advantage functions.

# 4 Experiment

## 4.1 Experimental Settings

To comprehensively evaluate the performance of SADN, we compare it with other RL methods that consider sequential actions, including ACE [19] and SAQL [3], as well as a range of general advanced MARL algorithms, including value-based methods like VDN [39] and QMIX [31], and policy gradient-based methods like MAPPO [46], HAPPO [17] and HASAC [22]. We also compare it with an extension of MAPPO, MAPPOar [11], which considers sequential actions by simply adding the prior actions to the state of the subsequent agents. We also investigate HAPPO's ability of sequential modeling by adjusting the permutation of updating agents.

We consider the following environments for comparing these methods, where the details of these environments can be found in Appendix B.

**Sigmoid.** The Sigmoid task [2] is basically an approximation task with hyperparameters changing over time. For a sampled instance $i$ and the $h$-th hyperparameter, the target sigmoid function to approximate is defined as $\text{sig}(t, s_{i,h}, p_{i,h}) = \frac{1}{1+e^{-s_{i,h}(t-p_{i,h})}}$, and the team reward at timestep $t$ on instance $i$ is defined as $r_t^i = \prod_{h=0}^{H-1}(1 - |\text{sig}(t, s_{i,h}, p_{i,h}) - a_{h,t}|)$, where $H$ is the number of hyperparameters, and $a_{h,t}$ is the configuration for the $h$-th hyperparameter at timestep $t$.

**Seq-Sigmoid.** Despite considering multiple hyperparameters, the original Sigmoid lacks hyperparameters with inherent inter-dependencies, making it unable to reflect the actual situation and the behavior of complex algorithms. Therefore, we modify the original Sigmoid into Seq-Sigmoid to bring in the inherent inter-dependencies among hyperparameters. The main modification is that at timestep $t$, the former hyperparameter's value will determine a scaling factor $\alpha_{h,t}$, which controls the latter hyperparameter's slope $s_{i,h}$. In this way, the former hyperparameter's configuration will have a strong influence on the configuration of the latter hyperparameter. Moreover, in order to avoid that having this dependency affects the configuration of the former hyperparameter itself, we use the formula $\min(|\text{sig}(t, \alpha_{h,t} \cdot s_{i,h}, p_{i,h}) - a_{h,t}|, |1 - \text{sig}(t, \alpha_{h,t} \cdot s_{i,h}, p_{i,h}) - a_{h,t}|)$ to measure the approximation, which returns the same value when two values of $a_{h,t}$ are symmetrical about 0.5. The pseudo-code of the Seq-Sigmoid benchmark is provided in Algorithm 1.

---
**Algorithm 1** Benchmark Outline: Seq-Sigmoid
---
1: **Benchmark Parameters:** number $H$ of hyperparameters, number $C_h$ of choice for each hyperparameter $h$, episode length $T$;
2: $s_i \sim \mathcal{U}(-100, 100, H)$;
3: $p_i \sim \mathcal{N}(T/2, T/4, H)$;
4: **for** $t \in \{0, 1, \dots, T\}$ **do**
5:     The state at step $t$: $\text{state}_t = s_i \cup p_i \cup \{t\}$;
6:     actions: $a_{h,t} \in \left\{\frac{0}{C_h}, \frac{1}{C_h}, \dots, \frac{C_h-1}{C_h}\right\}$ for all $0 \le h < H$ and $0 \le t \le T$;
7:     $\alpha_{0,t} = 1, \alpha_{h,t} = \begin{cases} 10 & \text{if } a_{h-1,t} \ge 0.5 \\ 0.1 & \text{if } a_{h-1,t} < 0.5 \end{cases}, 0 < h < H$;
8:     $r_t^i \leftarrow \prod_{h=0}^{H-1}(1 - \min(|\text{sig}(t, \alpha_{h,t} \cdot s_{i,h}, p_{i,h}) - a_{h,t}|, |1 - \text{sig}(t, \alpha_{h,t} \cdot s_{i,h}, p_{i,h}) - a_{h,t}|))$
9: **end for**
---

**Seq-Sigmoid-Mask.** This benchmark masks $s_{i,h}$ and $p_{i,h}$ (i.e., the agents can only observe $\text{state}_t = \{t\}$) and sets $s_{i,h} = 1$ to avoid too much randomness in Seq-Sigmoid, which makes the task a single instance task. That is, we denote the instance as $i_0$, and the goal is to approximate to $\text{sig}(t, 1, p_{i_0,h})$, but with randomness (i.e., $p_{i_0,h} \sim \mathcal{N}(T/2, T/4, H)$). This is to simulate the highly-random execution

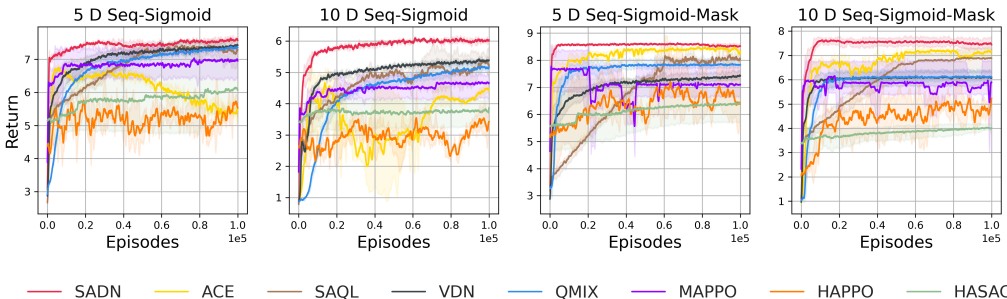

Figure 2: Training curves of return value obtained by the compared methods on four Seq-Sigmoid variant tasks, where the results are averaged over 6 runs.

of evolutionary algorithms, serving as a testbed for evaluating the DAC algorithms' ability when coping with randomness.

**Seq-Sigmoid-Robust** ($n$). This benchmark gives $n$ hyperparameters completely random configurations (regardless of how agents tune these $n$ hyperparameters), while all other hyperparameters are configured based on the actions. This is to simulate the complex scenarios where some agents can not learn the proper actions, serving as a testbed for evaluating the robustness of DAC algorithms.

**MOEA/D.** MOEA/D [44, 48] is a challenging task [44] based on the well-known multi-objective evolutionary algorithm MOEA/D with heterogeneous hyperparameters, which is a strong stochastic benchmark due to the randomness of the evolutionary optimization process. The action space of the MOEA/D environment includes four hyperparameters of MOEA/D: weights, neighborhood size, types of the reproduction operator, and the corresponding hyperparameters of the reproduction operators. The state includes the features of the problem instance, the optimization process, and the evolution statistics of the current population. For the reward, we use the triangle-based reward function proposed in [44], which has been proven effective in the MOEA/D environment.

### 4.2 RQ1: Is the sequential information useful?

On the original Sigmoid benchmark, we traditionally train the agents with a set of given instances (i.e., a set of $s_{i,h}$ and $p_{i,h}$) and test them on other sampled instances. In order to better reflect the generalization ability of the algorithms, we resample the instance (i.e., resample a new $s_{i,h}$ and $p_{i,h}$) every time one episode is done and the environment is reset. Therefore, the agents are truly trained over the distribution of the problem instances, and every point on the training curve can reflect the generalization ability of the learned policy at the exact training phase.

Figure 2 demonstrates the smoothed training curves of the return values of our method and the compared methods on Seq-Sigmoid and Seq-Sigmoid-Mask with 5/10 dimensions (i.e., 5/10 hyperparameters). As mentioned in the previous paragraph, we resample the instance every episode, and thereby the training curves reflect the policies' performance over the instance distribution at a certain training phase. It can be observed that our method, SADN, outperforms other methods on every benchmark, with faster convergence, less variance, and better final performance. Basically, the methods exploiting the sequential information (i.e., SADN, ACE, and SAQL) succeed in capturing the inherent inter-dependencies between the hyperparameters to some extent and achieve better performance, especially on the Seq-Sigmoid-Mask benchmark, where the traditional MARL methods fail to cope with the complex inter-dependencies and randomness. However, ACE shows a clear performance decay on the Seq-Sigmoid benchmark. This phenomenon may stem from the disruption of the long-chain action-value function update scheme in ACE by diverse problem instances, where certain action-value functions in the chain fail to achieve adaptation to novel instances.

The comparison of the sequential methods based on the correct order and the reverse order is given in Appendix C.2 due to space limitation. Basically, the reverse order underperforms the correct order

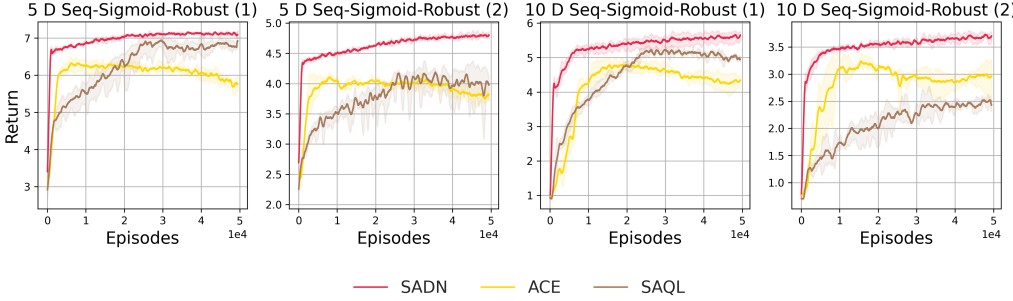

Figure 3: Training curves of return value obtained by the sequential methods on four Seq-Sigmoid-Robust tasks, where the results are averaged over 6 runs.

and produces a larger variance due to the wrong capture of the inherent inter-dependencies between the hyperparameters, except for HAPPO, which only considers the update order of the agents rather than the order of taking actions. The comparison of using the correct and the reverse order provides evidence for the effectiveness of modeling the inherent inter-dependencies correctly.

We also provide experimental results of our method and compared methods on the original Sigmoid in Appendix C.1. The results show that the effectiveness of our method even without strong inter-dependencies between the hyperparameters.

### 4.3 RQ2: How robust is SADN?

In complex dynamic parameter configuration problems, some agents often struggle to learn effectively, e.g., the configuration range of one certain parameter is not suitable, causing the corresponding agent to struggle to learn effectively. Especially in sequential scenarios, this issue can be more pronounced because those hindering agents can give misleading information through communication to other agents. Therefore, we test the methods designed for sequential decision-making on Seq-Sigmoid-Robust to examine their ability to cope with this issue.

Figure 3 plots the training curves of SADN (ours), ACE and SAQL on Seq-Sigmoid-Robust with 5/10 dimensions and 1/2 random agents. SADN shows better performance and robustness when coping with this complex situation, while ACE shows apparent performance decay because its fragile long-chain update scheme suffers from the instability of learning caused by the random agents, and SAQL sticks in poor performance due to the interference of the random agents.

### 4.4 RQ3: How does SADN perform on tuning real complex algorithms like MOEA/D?

In the MOEA/D environment, we set the proper parameter order as: 1) determining the neighborhood size to decide whether to explore or exploit; 2) choosing the operator type; 3) choosing the corresponding operator's parameters; 4) deciding whether to update the weights to bring in new subproblems. We compare our SADN with different state-of-the-art MARL methods, as well as the RL methods that consider sequential actions. We use the Inverted Generational Distance (IGD) [4] as the metric to measure the performance of the algorithms, which is the smaller the better.

As demonstrated in Table 1, the top three problems are the problems for training, and the bottom five problems are for testing. Our proposed SADN achieves significantly superior performance on most problem instances, and averagely ranks the best on all problem dimension settings. There are only occasional cases where SADN underperforms the baseline, and all of them occur in the testing phase. This may be owing to the very different function landscape that was not included in the training phase, and the learned RL policy fails to generalize. This issue is always of concern regarding the generalization ability of RL, and we believe good state and reward designs that capture commonalities among various problem sets can further enhance generalization, as well as more diverse training datasets. In general, our SADN shows strong generalization abilities across problem classes. The comparison results of using the correct and the reverse order, which are provided in Appendix C.3 due

Table 1: IGD values obtained by SADN and the compared methods on different problems. Each result consists of the mean of 10 runs. The best mean value on each problem is highlighted in bold. The symbols '+', '−' and '≈' indicate that the result is significantly superior to, inferior to, and almost equivalent to our method SADN, respectively, according to the Wilcoxon rank-sum test with significance level 0.05.

| Problem | Dim | MOEA/D [48] | SADN | ACE [19] | SAQL [3] | VDN [39] | QMIX [31] | MAPPO [46] | HAPPO [17] | HASAC [22] |
|---|---|---|---|---|---|---|---|---|---|---|
| DTLZ2 | 6 | 4.593e-02 | 3.809e-02 | 3.851e-02 | 3.950e-02 | 3.905e-02 | 3.906e-02 | **3.790e-02** | 3.838e-02 | 4.045e-02 |
| | 9 | 4.598e-02 | **3.875e-02** | 4.057e-02 | 4.337e-02 | 4.009e-02 | 4.120e-02 | 3.907e-02 | 4.095e-02 | 4.103e-02 |
| | 12 | 4.599e-02 | **3.874e-02** | 4.063e-02 | 4.698e-02 | 3.958e-02 | 4.161e-02 | 3.970e-02 | 4.075e-02 | 4.140e-02 |
| WFG4 | 6 | 5.729e-02 | **4.537e-02** | 4.626e-02 | 4.817e-02 | 4.904e-02 | 4.827e-02 | 4.639e-02 | 4.956e-02 | 5.703e-02 |
| | 9 | 5.736e-02 | **5.190e-02** | 5.853e-02 | 5.365e-02 | 5.800e-02 | 5.541e-02 | 5.341e-02 | 6.138e-02 | 6.169e-02 |
| | 12 | 5.751e-02 | **5.213e-02** | 5.751e-02 | 6.241e-02 | 5.687e-02 | 5.835e-02 | 5.340e-02 | 6.071e-02 | 6.187e-02 |
| WFG6 | 6 | 5.855e-02 | **3.908e-02** | 3.962e-02 | 3.964e-02 | 4.080e-02 | 4.214e-02 | 3.937e-02 | 4.022e-02 | 4.536e-02 |
| | 9 | 6.641e-02 | 4.884e-02 | 5.111e-02 | **4.384e-02** | 5.558e-02 | 7.416e-02 | 6.194e-02 | 5.921e-02 | 5.879e-02 |
| | 12 | 6.864e-02 | **4.837e-02** | 5.490e-02 | 5.676e-02 | 5.444e-02 | 7.999e-02 | 6.067e-02 | 5.218e-02 | 4.993e-02 |
| Training: +/−/≈ | | 0/9/0 | | 0/8/1 | 1/8/0 | 0/9/0 | 0/9/0 | 1/8/0 | 0/9/0 | 0/8/1 |
| DTLZ4 | 6 | 5.455e-02 | **3.895e-02** | 4.221e-02 | 5.019e-02 | 3.986e-02 | 4.331e-02 | 3.909e-02 | 3.987e-02 | 4.410e-02 |
| | 9 | 6.229e-02 | 7.319e-02 | 4.492e-02 | 6.895e-02 | **4.361e-02** | 7.108e-02 | 6.168e-02 | 4.645e-02 | 4.678e-02 |
| | 12 | 6.790e-02 | 7.852e-02 | 4.650e-02 | 8.589e-02 | **4.647e-02** | 5.532e-02 | 5.794e-02 | 5.107e-02 | 4.825e-02 |
| WFG5 | 6 | 6.303e-02 | 4.749e-02 | 4.752e-02 | 5.390e-02 | 4.761e-02 | 4.786e-02 | **4.719e-02** | 4.746e-02 | 4.744e-02 |
| | 9 | 6.351e-02 | 4.773e-02 | 4.792e-02 | 4.788e-02 | 4.784e-02 | 4.770e-02 | **4.762e-02** | 4.793e-02 | 4.780e-02 |
| | 12 | 6.381e-02 | 4.793e-02 | 4.791e-02 | 4.954e-02 | 4.780e-02 | **4.778e-02** | 4.789e-02 | 4.799e-02 | 4.785e-02 |
| WFG7 | 6 | 5.803e-02 | **3.893e-02** | 3.942e-02 | 3.958e-02 | 4.051e-02 | 4.207e-02 | 3.918e-02 | 3.980e-02 | 4.372e-02 |
| | 9 | 5.812e-02 | **4.053e-02** | 4.470e-02 | 4.257e-02 | 4.566e-02 | 4.119e-02 | 4.607e-02 | 4.632e-02 | 4.689e-02 |
| | 12 | 5.814e-02 | **4.060e-02** | 4.447e-02 | 4.975e-02 | 4.432e-02 | 4.445e-02 | 4.132e-02 | 4.597e-02 | 4.689e-02 |
| WFG8 | 6 | **7.875e-02** | 1.018e-01 | 1.029e-01 | 1.054e-01 | 1.169e-01 | 1.054e-01 | 1.037e-01 | 1.156e-01 | 1.252e-01 |
| | 9 | 9.680e-02 | **7.853e-02** | 9.787e-02 | 9.716e-02 | 8.720e-02 | 8.180e-02 | 7.880e-02 | 9.964e-02 | 9.991e-02 |
| | 12 | 8.710e-02 | **7.891e-02** | 8.387e-02 | 9.023e-02 | 8.552e-02 | 8.279e-02 | 7.923e-02 | 8.636e-02 | 8.734e-02 |
| WFG9 | 6 | 5.600e-02 | 3.987e-02 | 4.012e-02 | 4.010e-02 | 4.156e-02 | 4.266e-02 | **3.986e-02** | 4.041e-02 | 4.395e-02 |
| | 9 | 5.748e-02 | 4.341e-02 | 4.552e-02 | **4.281e-02** | 7.965e-02 | 5.274e-02 | 4.573e-02 | 5.079e-02 | 5.429e-02 |
| | 12 | 5.827e-02 | **4.249e-02** | 4.477e-02 | 7.291e-02 | 8.023e-02 | 5.246e-02 | 4.495e-02 | 6.556e-02 | 8.189e-02 |
| Testing: +/−/≈ | | 1/12/2 | | 2/9/4 | 0/11/4 | 2/12/1 | 2/11/2 | 4/5/6 | 2/11/2 | 2/10/3 |
| average rank | 6 | 8 | **1.75** | 3.625 | 5.5 | 5.75 | 6.25 | 2 | 4.875 | 7.25 |
| | 9 | 7.375 | **2.5** | 4.75 | 4.375 | 4.75 | 5.375 | 3.125 | 6.25 | 6.5 |
| | 12 | 7.25 | **2.5** | 3.875 | 8.125 | 3.5 | 4.875 | 3.625 | 5.5 | 5.75 |

to space limitation, also demonstrate the effectiveness of correctly considering the parameter order on the real-scenario multi-objective evolutionary algorithm. Generally, the correct order produces better results than the reverse order within the same sequential methods.

Moreover, we also provide a discussion of the generalization ability across different dimensions in Appendix C.4. Overall, the results demonstrate that our proposed method exhibits good generalization abilities across different dimensions. However, cross-dimensional generalization remains a significant challenge and is worth further investigation, such as through multi-dimensional mixed training and employing multi-head models.

## 5 Conclusion

This paper considers the inherent inter-dependencies among hyperparameters in DAC, which are natural and common in practice. We propose a contextual sequential MMDP formulation to capture the inherent inter-dependencies among hyperparameters and also propose a sequential advantage decomposition network to solve it. Experiments from synthetic white-box tasks with apparent inter-dependencies to a complex real-scenario multi-objective evolutionary algorithm demonstrate the effectiveness of our proposed method, as well as its strong generalization ability. One important future work is to modularize the algorithms that support more flexible reconstruction of different algorithm sub-modules to design diverse algorithm structures [12].

One limitation is that the hyperparameter order requires to be given in advance. In practice, there is an intuitive and natural way to determine the order: setting it according to the sequence in which the hyperparameters take effect during the execution of the algorithm. Specifically, in the code implementation of the target algorithm, all hyperparameters take effect in a specific order, which is the order we have set. This strategy aligns with our intuition and performs well in the experiments of this paper. In the future, one could apply large language models to set the proper order of the hyperparameter configuration, as well as causal models to automatically learn the hyperparameter's causal structure from the data [25].

## Acknowledgment

The authors thank anonymous reviewers for their insightful and valuable comments. This work was supported by the National Science and Technology Major Project (2022ZD0116600), the National Science Foundation of China (62276124, 624B2069, 62506159), the Fundamental Research Funds for the Central Universities (14380020), and the Young Elite Scientists Sponsorship Program by CAST for PhD Students. The authors want to acknowledge support from the Huawei Technology Cooperation Project.

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

# A  Proof

## A.1  Deriving the global advantage update scheme from the Q-learning update scheme

In this appendix, we derive our global advantage update scheme (i.e., the GAE with $\lambda = 0$):

$$A(s, \boldsymbol{a}) \leftarrow A(s, \boldsymbol{a}) + \alpha[r + \gamma V(s') - V(s) - A(s, \boldsymbol{a})]$$

from the Q-learning update scheme

$$Q(s, \boldsymbol{a}) \leftarrow Q(s, \boldsymbol{a}) + \alpha[r + \gamma \max_{\boldsymbol{a}'} Q(s', \boldsymbol{a}') - Q(s, \boldsymbol{a})]$$

as follows:

$$Q(s, \boldsymbol{a}) \leftarrow Q(s, \boldsymbol{a}) + \alpha[r + \gamma \max_{\boldsymbol{a}'} Q(s', \boldsymbol{a}') - Q(s, \boldsymbol{a})]$$

$$A(s, \boldsymbol{a}) + V(s) \leftarrow A(s, \boldsymbol{a}) + V(s) + \alpha[r + \gamma \max_{\boldsymbol{a}'}(A(s', \boldsymbol{a}') + V(s')) - A(s, \boldsymbol{a}) - V(s)]$$

$$A(s, \boldsymbol{a}) \leftarrow A(s, \boldsymbol{a}) + \alpha[r + \gamma \max_{\boldsymbol{a}'} A(s', \boldsymbol{a}') + \gamma V(s') - A(s, \boldsymbol{a}) - V(s)]$$

$$A(s, \boldsymbol{a}) \leftarrow A(s, \boldsymbol{a}) + \alpha[r + \gamma V(s') - V(s) - A(s, \boldsymbol{a})]$$

Note that in Q-learning, we choose the action greedily with respect to the learned Q-function by $\pi(s) := \arg\max_{a \in \mathcal{A}} Q(s, a)$, and $V(s) = \mathbb{E}_{\pi}[\sum_{t=0}^{\infty} \gamma^t r_t | s_0 = s] = \max_{a \in \mathcal{A}} Q(s, a)$. Therefore, $\max_{\boldsymbol{a}'} A(s', \boldsymbol{a}') = \max_{\boldsymbol{a}'} Q(s', \boldsymbol{a}') - V(s') = 0$.

## A.2  Proof of Theorem 1

**Theorem 1.** *Selecting the best joint action $\boldsymbol{a}$ for state $s$ to maximize the global action value function is equivalent to sequentially selecting the best action $a_i$ for state $s$ to maximize the agent $i$'s advantage function. That is,*

$$\arg\max_{\boldsymbol{a}} Q(s, \boldsymbol{a}) = \begin{pmatrix} \arg\max_{a_1} A_1(s, a_1) \\ \arg\max_{a_2} A_2(s, \arg\max_{a_1} A_1(s, a_1), a_2) \\ \vdots \\ \arg\max_{a_n} A_n(s, \arg\max_{a_1} A_1(s, a_1), \arg\max_{a_2} A_2(s, \arg\max_{a_1} A_1(s, a_1), a_2), \dots, a_n) \end{pmatrix}.$$

*Proof.* Let $\boldsymbol{a}^*$ denote the optimal action that maximizes the Q-function (that is, $\boldsymbol{a}^* = \arg\max_{\boldsymbol{a}} Q(s, \boldsymbol{a})$), and for the agent $i$, its policy based on the actions of its prior agents be denoted as $\pi_i(s, \boldsymbol{a}_{1:i-1}) := \arg\max_{a_i} A_i(s, \boldsymbol{a}_{1:i-1}, a_i)$ and the joint policy $\boldsymbol{\pi} := (\pi_1, \pi_2, \dots, \pi_n)$.

By induction, we prove that for the agent $i$, if its prior agents have selected the optimal actions $\boldsymbol{a}_{1:i-1}^*$, it can select its optimal action $a_i^*$ by maximizing its advantage function $A_i(s, \boldsymbol{a}_{1:i-1}^*, a_i)$.

For agent $n$:

$$\begin{aligned} a_n^* &= \arg\max_{a_n} Q(s, \boldsymbol{a}_{1:n-1}^*, a_n) \\ &= \arg\max_{a_n}[Q(s, \boldsymbol{a}_{1:n-1}^*, a_n) - \mathbb{E}_{a_n \sim \pi_n}[Q(s, \boldsymbol{a}_{1:n-1}^*, a_n)]] \\ &= \arg\max_{a_n} A_n(s, \boldsymbol{a}_{1:n-1}^*, a_n) \end{aligned}$$

where the second equation holds because $\mathbb{E}_{a_n \sim \pi_n}\left[Q(s, \boldsymbol{a}_{1:n-1}^*, a_n)\right]$ is a constant, and the third equation refers to the definition of $A_n(s, \boldsymbol{a}_{1:n-1}^*, a_n)$ in Definition 2.

We now assume that for the agent $i+1$ to $n$, this property holds, and for the agent $i$:

$$\begin{aligned} a_i^* &= \arg\max_{a_i} Q(s, \boldsymbol{a}_{1:i-1}^*, a_i, \arg\max_{\boldsymbol{a}_{i+1:n}} Q(s, \boldsymbol{a}_{1:i-1}^*, a_i, \boldsymbol{a}_{i+1:n})) \\ &= \arg\max_{a_i} \mathbb{E}_{\boldsymbol{a}_{i+1:n} \sim \boldsymbol{\pi}_{i+1:n}}[Q(s, \boldsymbol{a}_{1:i-1}^*, a_i, \boldsymbol{a}_{i+1:n})] \\ &= \arg\max_{a_i}\{\mathbb{E}_{\boldsymbol{a}_{i+1:n} \sim \boldsymbol{\pi}_{i+1:n}}[Q(s, \boldsymbol{a}_{1:i-1}^*, a_i, \boldsymbol{a}_{i+1:n})] - \mathbb{E}_{\boldsymbol{a}_{i:n} \sim \boldsymbol{\pi}_{i:n}}[Q(s, \boldsymbol{a}_{1:i-1}^*, \boldsymbol{a}_{i:n})]\} \\ &= \arg\max_{a_i} A_i(s, \boldsymbol{a}_{1:i-1}^*, a_i) \end{aligned}$$

where the second equation holds because the inductive hypothesis implies that if the $i$-th agent selects the optimal $a_i^*$, $\boldsymbol{\pi}_{i+1:n}(s, \boldsymbol{a}_{1:i}^*) = \boldsymbol{a}_{i+1:n}^*$, and thus we have

$$\forall a_i, \boldsymbol{a}_{i+1:n}, \mathbb{E}_{\boldsymbol{a}_{i+1:n} \sim \boldsymbol{\pi}_{i+1:n}}[Q(s, \boldsymbol{a}_{1:i-1}^*, a_i^*, \boldsymbol{a}_{i+1:n})] = Q(s, \boldsymbol{a}_{1:i-1}^*, a_i^*, \boldsymbol{a}_{i+1:n}^*)$$
$$\geq Q(s, \boldsymbol{a}_{1:i-1}^*, a_i, \boldsymbol{a}_{i+1:n})$$

which means

$$\forall a_i, \mathbb{E}_{\boldsymbol{a}_{i+1:n} \sim \boldsymbol{\pi}_{i+1:n}}[Q(s, \boldsymbol{a}_{1:i-1}^*, a_i^*, \boldsymbol{a}_{i+1:n})] \geq \mathbb{E}_{\boldsymbol{a}_{i+1:n} \sim \boldsymbol{\pi}_{i+1:n}}[Q(s, \boldsymbol{a}_{1:i-1}^*, a_i, \boldsymbol{a}_{i+1:n})]$$

Therefore, $a_i^* = \arg\max_{a_i} \mathbb{E}_{\boldsymbol{a}_{i+1:n} \sim \boldsymbol{\pi}_{i+1:n}}[Q(s, \boldsymbol{a}_{1:i-1}^*, a_i, \boldsymbol{a}_{i+1:n})]$, and the second equation holds. The third equation holds because $\mathbb{E}_{\boldsymbol{a}_{i:n} \sim \boldsymbol{\pi}_{i:n}}[Q(s, \boldsymbol{a}_{1:i-1}^*, \boldsymbol{a}_{i:n})]$ is a constant, and the forth equation refers to the definition of $A_i(s, \boldsymbol{a}_{1:i-1}^*, a_i)$ in Definition 2.

Therefore, by induction, we conclude:

$$\forall i, a_i^* = \arg\max_{a_i} A_i(s, \boldsymbol{a}_{1:i-1}^*, a_i)$$

Notably, when $i = 1$, $a_1^* = \arg\max_{a_1} A_1(s, a_1)$, and for subsequent agents, the agent 2 will select $a_2^* = \arg\max_{a_2} A_2(s, a_1^*, a_2)$, and so on, with the agent $n$ selecting $a_n^* = \arg\max_{a_n} A_n(s, \boldsymbol{a}_{1:n-1}^*, a_n)$. $\qquad\square$

Thus, we have completed the proof of Theorem 1.

### A.3  Proof of Lemma 1

**Lemma 1** (Multi-agent Advantage Decomposition [17])**.** *In any cooperative Markov game, given a joint policy $\boldsymbol{\pi}$, the global advantage function $A^{\boldsymbol{\pi}}(s, \boldsymbol{a})$, and $n$ agents in total, for any state $s$, the following equations hold:*

$$A^{\boldsymbol{\pi}}(s, \boldsymbol{a}) = \sum_{i=1}^{n} A_i^{\boldsymbol{\pi}}(s, \boldsymbol{a}_{1:i-1}, a_i)$$

*Proof.* Similar to the proof in [17], according to the Definition 2, we have

$$A^{\boldsymbol{\pi}}(s, \boldsymbol{a}) = Q^{\boldsymbol{\pi}}(s, \boldsymbol{a}) - V^{\boldsymbol{\pi}}(s)$$
$$= \sum_{i=1}^{n} Q_{1:i}^{\boldsymbol{\pi}}(s, \boldsymbol{a}_{1:i}) - Q_{1:i-1}^{\boldsymbol{\pi}}(s, \boldsymbol{a}_{1:i-1})$$
$$= \sum_{i=1}^{n} A_i^{\boldsymbol{\pi}}(s, \boldsymbol{a}_{1:i-1}, a_i)$$

which finishes the proof. $\qquad\square$

## B  Detailed settings and information of the environments

In this section, we introduce our detailed experimental settings, including network architectures, experimental configurations, and hyperparameter settings.

### B.1  Seq-Sigmoid

In this subsection, we introduce the original Sigmoid benchmark, the Seq-Sigmoid benchmark, and its variants in detail.

**Sigmoid [2].**  The pseudo-code of the Sigmoid benchmark is provided in Algorithm 2. The Sigmoid task is basically an approximation task with parameters changing over time, with the approximation target as $\text{sig}(t, s_{i,h}, p_{i,h})$, the state provided in line 5, and the reward at $t$ calculated in line 7.

---

**Algorithm 2** Benchmark Outline: Sigmoid

---

1: **Benchmark Parameters:** number $H$ of hyperparameters, number $C_h$ of choice for each hyper-parameter $h$, episode length $T$;
2: $s_i \sim \mathcal{U}(-100, 100, H)$;
3: $p_i \sim \mathcal{N}(T/2, T/4, H)$;
4: **for** $t \in \{0, 1, \ldots, T\}$ **do**
5:     The state at step $t$: $\text{state}_t = s_i \cup p_i \cup \{t\}$;
6:     Select actions: $a_{h,t} \in \left\{ \frac{0}{C_h}, \frac{1}{C_h}, \ldots, \frac{C_h-1}{C_h} \right\}$ for all $0 \leq h < H$ and $0 \leq t \leq T$;
7:     $r_t^i \leftarrow \prod_{h=0}^{H-1} (1 - |\text{sig}(t, s_{i,h}, p_{i,h}) - a_{h,t}|)$;
8: **end for**

---

**Seq-Sigmoid.** The pseudo-code of the Seq-Sigmoid benchmark is provided in Algorithm 1. The main modification (i.e., in lines 7 and 8) is that, at timestep $t$, the former parameter's value will determine a scaling factor $\alpha_{h,t}$, which controls the latter parameter's slope $s_{i,h}$. In this way, the former parameter configuration will have a strong influence on the configuration of the latter parameter. Moreover, to avoid having this dependency affect the configuration of the former parameter itself, we use the formula $\min(|\text{sig}(t, \alpha_{h,t}s_{i,h}, p_{i,h}) - a_{h,t}|, |1 - \text{sig}(t, \alpha_{h,t}s_{i,h}, p_{i,h}) - a_{h,t}|)$ to measure the approximation, which returns the same value when two $a_{h,t}$ are symmetrical about 0.5.

**Seq-Sigmoid-Mask.** The pseudo-code of the Seq-Sigmoid-Mask benchmark is provided in Algorithm 3. The main modification compared to the Seq-Sigmoid benchmark is that we mask $s_{i,h}$ and $p_{i,h}$ (i.e., the $\text{state}_t \in \{t\}$ in line 4) and set $s_{i,h} = 1$ (i.e., in line 7) to avoid too much randomness.

---

**Algorithm 3** Benchmark Outline: Seq-Sigmoid-Mask

---

1: **Benchmark Parameters:** number $H$ of hyperparameters, number $C_h$ of choice for each hyper-parameter $h$, episode length $T$;
2: $p_i \sim \mathcal{N}(T/2, T/4, H)$;
3: **for** $t \in \{0, 1, \ldots, T\}$ **do**
4:     The state at step $t$: $\text{state}_t = \{t\}$;
5:     Select actions: $a_{h,t} \in \left\{ \frac{0}{C_h}, \frac{1}{C_h}, \ldots, \frac{C_h-1}{C_h} \right\}$ for all $0 \leq h < H$ and $0 \leq t \leq T$;
6:     $\alpha_{0,t} = 1, \alpha_{h,t} = \begin{cases} 10 & \text{if } a_{h-1,t} \geq 0.5 \\ 0.1 & \text{if } a_{h-1,t} < 0.5 \end{cases}, 0 < h < H$;
7:     $r_t^i \leftarrow \prod_{h=0}^{H-1} (1 - \min(|\text{sig}(t, \alpha_{h,t} \cdot 1, p_{i,h}) - a_{h,t}|, |1 - \text{sig}(t, \alpha_{h,t} \cdot 1, p_{i,h}) - a_{h,t}|))$;
8: **end for**

---

**Seq-Sigmoid-Robust ($n$).** The pseudo-code of the Seq-Sigmoid-Robust ($n$) benchmark is provided in Algorithm 4. The benchmark gives random configurations for the $n$ parameters, no matter how the corresponding agents tune them in Seq-Sigmoid (i.e., in line 7), with other parameters configured properly, to simulate the ineffective learning of some agents in complex dynamic algorithm configuration scenarios. In particular, we give random configurations for the $\lfloor n/2 \rfloor$-th parameter in Seq-Sigmoid-Robust (1) and the $\lfloor n/2 \rfloor$-th and the $\lfloor n/2 \rfloor + 1$-th parameters in Seq-Sigmoid-Robust (2).

## B.2 MOEA/D

MOEA/D is the proposed DAC benchmark [44] based on a real-scenario multi-objective evolutionary algorithm MOEA/D [48] with heterogeneous parameters, which is a strong stochastic benchmark due to the randomness of the evolution optimization process.

**MOEA/D algorithm** The multi-objective evolutionary algorithm MOEA/D is originally designed to solve the Multi-objective Optimization Problems (MOPs), which can be defined as

$$\min \ \boldsymbol{F}(\boldsymbol{x}) = (f_1(\boldsymbol{x}), \ldots, f_m(\boldsymbol{x})) \quad \text{s.t.} \quad \boldsymbol{x} \in \Omega,$$

where $\boldsymbol{x} = (x_1, \ldots, x_D)$ is a solution, $\boldsymbol{F} : \Omega \rightarrow \mathbb{R}^m$ constitutes $m$ objective functions, $\Omega = [x_i^L, x_i^U]^D \subseteq \mathbb{R}^D$ is the solution space, and $\mathbb{R}^m$ is the objective space. MOP aims to find a set of

---

**Algorithm 4** Benchmark Outline: Seq-Sigmoid-Robust (n)

---

1: **Benchmark Parameters:** number $H$ of hyperparameters, number $C_h$ of choice for each hyper-parameter $h$, episode length $T$, $n$ random hyperparameter indexes $h_1, h_2, \ldots, h_n$;

2: $s_i \sim \mathcal{U}(-100, 100, H)$;

3: $p_i \sim \mathcal{N}(T/2, T/4, H)$;

4: **for** $t \in \{0, 1, \ldots, T\}$ **do**

5:     The state at step $t$: $\text{state}_t \in s_i \cup p_i \cup \{t\}$;

6:     Actions: $a_{h,t} \in \left\{ \frac{0}{C_h}, \frac{1}{C_h}, \ldots, \frac{C_h-1}{C_h} \right\}$ for all $0 \leq h < H$ and $0 \leq t \leq T$;

7:     Reselect $a_{i,t}, i \in \{h_1, h_2, \ldots, h_n\}$ at random;

8:     $\alpha_{0,t} = 1, \alpha_{h,t} = \begin{cases} 10 & \text{if } a_{h-1,t} \geq 0.5 \\ 0.1 & \text{if } a_{h-1,t} < 0.5 \end{cases}, 0 < h < H$;

9:     $r_t^i \leftarrow \prod_{h=0}^{H-1} \left( 1 - \min(|\text{sig}(t, \alpha_{h,t} s_{i,h}, p_{i,h}) - a_{h,t}|, |1 - \text{sig}(t, \alpha_{h,t} s_{i,h}, p_{i,h}) - a_{h,t}|) \right)$;

10: **end for**

---

solutions that represent the best possible trade-offs between competing objectives. The objective vectors of these solutions are collectively called the Pareto front (PF), defined as follows:

**Definition 3.** *A solution $x^*$ is Pareto-optimal with respect to Eq. (B.2), if $\nexists x \in \Omega$ such that $\forall i : f_i(x) \leq f_i(x^*)$ and $\exists i : f_i(x) < f_i(x^*)$. The set of all Pareto-optimal solutions is called Pareto-optimal Set (PS). The set of the corresponding objective vectors of PS, i.e., $\{F(x) \mid x \in PS\}$, is called Pareto Front (PF).*

MOEA/D consists of two main processes, *decomposition* and *collaboration* [48, 41, 20]. In decomposition, MOEA/D solves a number of sub-problems through a number of weights and an aggregation function to approximate the PF. Several aggregation functions have been proposed for MOEA/D. Here, we introduce the common Tchebycheff approach (TCH) used in our paper as follows:

Given a weight vector $w = (w_1, \ldots, w_m)$ where $w_i \geq 0, \forall i \in \{1, \ldots, m\}$ and $\sum_{i=1}^{m} w_i = 1$, the sub-problem by TCH is formulated as

$$\min_{x \in \Omega} g(x \mid w, z^*) = \max_{1 \leq i \leq m} \{w_i \cdot |f_i(x) - z_i^*|\},$$

where $z^* = (z_1^*, \ldots, z_m^*)$ is the ideal point consisting of the best objective values obtained so far.

The fundamental intuition behind collaboration is that adjacent sub-problems tend to share similar properties. For instance, they may have similar objective functions and/or optimal solutions [20]. In particular, the neighborhood of a sub-problem is controlled by the Euclidean distance of its corresponding weight vector with respect to the others, as well as the hyperparameter *neighborhood size*: two sub-problems are neighbors of each other if the distance between them is smaller than the neighborhood size. In the selection process for a sub-problem, parent solutions are randomly selected from the corresponding neighborhood of each sub-problem. The sub-problem solutions within the same neighborhood are then replaced with the newly generated offspring solution if it is better than the current one.

The pseudo-code of MOEA/D is shown in Algorithm 5.

**Action** The action space of MOEA/D is of four dimensions, corresponding to the four heterogeneous MOEA/D parameters:

1. Weights. In MOEA/D, weights are used to transform an MOP into multiple single-objective sub-problems. Inspired by MOEA/D-AWA [29], we set the action space for weights by adjusting (T) or not adjusting (N) the weights. If the action is T, the weights will be updated; otherwise, the weights will remain the same. The weights adaptation mechanism is as follows.

The sparsity level of each solution $x^{(i)}$ is calculated based on vicinity distance [18]:

$$SL\left(x^{(i)}, \{x^{(p)}\}_{p=1}^N\right) = \prod_{j=1}^{m} l(x^{(i)}, j),$$

where $l(x^{(i)}, j)$ is the Euclidean distance between $x^{(i)}$ and its $j$-th nearest neighbor in the population $\{x^{(p)}\}_{p=1}^N$. In the calculation, the $m$ closest neighbors in the population are used, where $m$ denotes

---
**Algorithm 5** MOEA/D
---
**Input:** Population size $N$, number $T$ of iterations
**Output:** A set of Pareto-optimal solutions
  1: Initialize a population $\{\boldsymbol{x}^{(i)}\}_{i=1}^{N}$ of solutions, and a corresponding set $W = \{\boldsymbol{w}^{(i)}\}_{i=1}^{N}$ of weight vectors
  2: $t \leftarrow 0$
  3: **while** $t < T$ **do**
  4:     **for** $i = 1 : N$ **do**
  5:         Randomly select parent solutions from the neighborhood of $\boldsymbol{w}^{(i)}$, denoted as $\Theta^{\boldsymbol{w}^{(i)}}$
  6:         Use crossover and mutation operators to generate an offspring solution $\boldsymbol{x}'^{(i)}$
  7:         Evaluate the offspring solution to obtain $\boldsymbol{F}(\boldsymbol{x}'^{(i)})$
  8:         Update the ideal point $\boldsymbol{z}^*$:
  9:         **for** $j \in \{1, 2, \ldots, m\}$ **do**
10:            **if** $f_j(\boldsymbol{x}'^{(i)}) < z_j^*$ **then**
11:               $z_j^* \leftarrow f_j(\boldsymbol{x}'^{(i)})$
12:            **end if**
13:         **end for**
14:         Update the corresponding solution of each sub-problem within $\Theta^{\boldsymbol{w}^{(i)}}$ by $\boldsymbol{x}'^{(i)}$:
15:         **for** $\boldsymbol{w}^{(j)} \in \Theta^{\boldsymbol{w}^{(i)}}$ **do**
16:            **if** $g(\boldsymbol{x}'^{(i)} \mid \boldsymbol{w}^{(j)}, \boldsymbol{z}^*) < g(\boldsymbol{x}^{(j)} \mid \boldsymbol{w}^{(j)}, \boldsymbol{z}^*)$ **then**
17:               $\boldsymbol{x}^{(j)} \leftarrow \boldsymbol{x}'^{(i)}$
18:            **end if**
19:         **end for**
20:     **end for**
21:     $t \leftarrow t + 1$
22: **end while**
---

the number of objectives. Sub-problems corresponding to the solutions with sparsity levels in the bottom 5%, namely the overcrowded solutions, are subsequently removed.

To ensure that the total number of the sub-problems is still $N$, $0.05N$ new sub-problems and their corresponding solutions should be added, which come from an elite population that keeps all historical non-dominated solutions, with a maximum capacity of $1.5N$. When the size of the elite population surpasses this capacity, the solutions with the lowest sparsity level are eliminated. For every solution $\boldsymbol{x}'$ in the elite population, its sparsity level is calculated with respect to the current population (i.e., $SL(\boldsymbol{x}', \text{Pop})$, where Pop represents the set of the remaining $0.95N$ solutions). Subsequently, the solution with the highest sparsity level with respect to the current population is selected from the elite population and added to the current population. This selection and addition procedure is carried out for $0.05N$ times. For each newly added solution, the corresponding sub-problem (i.e., weight vector) is generated in a specific way as Algorithm 3 in [29].

2. Neighborhood size. The neighborhood size is used to control the maximum distance between solutions and their neighbors. A small size aims to exploit the local area, while a large size aims to explore a wide objective space [43]. The action space is of four dimensions, that is, 15, 20, 25 and 30, where 20 is the default value.

3. Types of reproduction operators. We consider four types of DE operators with different search abilities introduced in [21]. For reproducing an offspring solution for the $i$-th sub-problem, let $\boldsymbol{x}^{(i)}$ and $\boldsymbol{x}'^{(i)}$ denote its current solution and the generated offspring solution, respectively, and the equations for four types of DE operators are shown as follows:

- OP1: $\boldsymbol{x}'^{(i)} = \boldsymbol{x}^{(i)} + F \times \left( \boldsymbol{x}^{(r_1)} - \boldsymbol{x}^{(r_2)} \right),$

- OP2: $\boldsymbol{x}'^{(i)} = \boldsymbol{x}^{(i)} + F \times \left( \boldsymbol{x}^{(r_1)} - \boldsymbol{x}^{(r_2)} \right) + F \times \left( \boldsymbol{x}^{(r_3)} - \boldsymbol{x}^{(r_4)} \right),$

- OP3: $\boldsymbol{x}'^{(i)} = \boldsymbol{x}^{(i)} + K \times \left( \boldsymbol{x}^{(i)} - \boldsymbol{x}^{(r_1)} \right) + F \times \left( \boldsymbol{x}^{(r_2)} - \boldsymbol{x}^{(r_3)} \right) + F \times \left( \boldsymbol{x}^{(r_4)} - \boldsymbol{x}^{(r_5)} \right),$

- OP4: $\boldsymbol{x}'^{(i)} = \boldsymbol{x}^{(i)} + K \times \left( \boldsymbol{x}^{(i)} - \boldsymbol{x}^{(r_1)} \right) + F \times \left( \boldsymbol{x}^{(r_2)} - \boldsymbol{x}^{(r_3)} \right).$

Table 2: State at step $t$ in MOEA/D.

| Index | Parts of state | Feature | Notes |
|---|---|---|---|
| 0 | 1 | $1/m$ | $m$: Number of objectives |
| 1 | 1 | $1/D$ | $D$: Number of variables |
| 2 | 2 | $t/T$ | Computational budget that has been used |
| 3 | 2 | $N_{\text{stag}}/T$ | Stagnant count ratio |
| 4 | 3 | $\text{HV}_t$ | Hypervolume value |
| 5 | 3 | $\text{NDRatio}_t$ | Ratio of non-dominated solutions |
| 6 | 3 | $\text{Dist}_t$ | Average distance |
| 7 | 3 | $\text{HV}_t - \text{HV}_{t-1}$ | Change of HV between steps $t$ and $t-1$ |
| 8 | 3 | $\text{NDRatio}_t - \text{NDRatio}_{t-1}$ | Change of NDRatio between steps $t$ and $t-1$ |
| 9 | 3 | $\text{Dist}_t - \text{Dist}_{t-1}$ | Change of Dist between steps $t$ and $t-1$ |
| 10 | 3 | $\text{Mean}(\text{List}(\text{HV}, t, 5))$ | Mean of HV in the last 5 steps |
| 11 | 3 | $\text{Mean}(\text{List}(\text{NDRatio}, t, 5))$ | Mean of NDRatio in the last 5 steps |
| 12 | 3 | $\text{Mean}(\text{List}(\text{Dist}, t, 5))$ | Mean of Dist in the last 5 steps |
| 13 | 3 | $\text{Std}(\text{List}(\text{HV}, t, 5))$ | Standard deviation of HV in the last 5 steps |
| 14 | 3 | $\text{Std}(\text{List}(\text{NDRatio}, t, 5))$ | Standard deviation of NDRatio in the last 5 steps |
| 15 | 3 | $\text{Std}(\text{List}(\text{Dist}, t, 5))$ | Standard deviation of Dist in the last 5 steps |
| 16 | 3 | $\text{Mean}(\text{List}(\text{HV}, t, t))$ | Mean of HV in all the steps so far |
| 17 | 3 | $\text{Mean}(\text{List}(\text{NDRatio}, t, t))$ | Mean of NDRatio in all the steps so far |
| 18 | 3 | $\text{Mean}(\text{List}(\text{Dist}, t, t))$ | Mean of Dist in all the steps so far |
| 19 | 3 | $\text{Std}(\text{List}(\text{HV}, t, t))$ | Standard deviation of HV in all the steps so far |
| 20 | 3 | $\text{Std}(\text{List}(\text{NDRatio}, t, t))$ | Standard deviation of NDRatio in all the steps so far |
| 21 | 3 | $\text{Std}(\text{List}(\text{Dist}, t, t))$ | Standard deviation of Dist in all the steps so far |

Here, $\boldsymbol{x}^{(r_1)}, \boldsymbol{x}^{(r_2)}, \boldsymbol{x}^{(r_3)}, \boldsymbol{x}^{(r_4)}$, and $\boldsymbol{x}^{(r_5)}$ are different parent solutions randomly selected from the neighborhood of $\boldsymbol{x}^{(i)}$. The scaling factor $F > 0$ controls the impact of the vector differences on the mutant vector, and $K \in [0, 1]$ functions similarly to $F$.

4. Parameters of reproduction operators. The parameters (particularly the scaling factor) of the reproduction operators in MOEA/D have a significant effect on the algorithm's performance [38]. We set the scaling factor $K$ to a fixed value of 0.5 as recommended [21], and dynamically adjust the scaling factor $F$ with four discrete dimensions, i.e., 0.4, 0.5, 0.6 and 0.7, with 0.5 serving as the default value.

**State** The state of MOEA/D includes three parts: 1. The feature of the problem instance (e.g., the numbers of objectives and variables); 2. The feature of the optimization process (e.g., the budget has been used); 3. Information about the evolution situation of the population (e.g., the hypervolume value and the average distance of the current population). The detailed state feature is demonstrated in Table 2.

**Transition** One step of transition takes place at one generation change in the evolutionary process of MOEA/D.

**Reward** A positive reward of the MOEA/D environment is assigned to the agent team when a better solution is found compared to the current best solution. Since optimization becomes harder when the current best solution is more optimal with time, we should assign more reward to better solutions found in the latter stage. We use the triangle-based reward function proposed in [44] defined as follows, which has been proved effective in the MOEA/D environment.

At step $t$,

$$r_t = \begin{cases} (1/2) \cdot (p_{t+1}^2 - p_t^2) & \text{if } f(s_{t+1}) < f_t^* \\ 0 & \text{otherwise} \end{cases},$$

where

$$p_{t+1} = \begin{cases} \frac{f(s_0) - f(s_{t+1})}{f(s_0)} & \text{if } f(s_{t+1}) < f_t^* \\ p_t & \text{otherwise} \end{cases},$$

and $f_t^*$ is the minimum metric value found until step $t$.

**Problem instance**   The problem instances of MOEA/D include the well-known multi-objective optimization problems (MOP) benchmarks DTLZ [8] and WFG [13] with variable numbers of objectives and problem dimensions, which cover different difficulty levels of MOPs.

### B.3   Hyperparameters of the algorithms compared in the experiments

For all the compared MARL algorithms, we use their default suggested hyperparameter settings in EPyMARL[2] (i.e., VDN [39], QMIX [31], MAPPO [46]) or their official implementation (i.e., HASAC [22]). For a fair comparison, we use the same hyperparameters for algorithms of the same type (i.e., we use the same hyperparameters for the value-based methods: SADN, ACE [19], SAQL [3], VDN [39] and QMIX [31], as well as the same hyperparameters for the policy gradient-based PPO methods: MAPPO [46] and HAPPO [17]). The detailed hyperparameters in the experiments are given in Table 3. For a fair comparison, we use one single 64-dimensional hidden layer for all the value networks and the policy networks.

Table 3: The hyperparameters of the compared algorithms in the MOEA/D environment, and "-" means the certain algorithm does not have that hyperparameter.

| Hyperparameter | SADN | ACE | SAQL | VDN | QMIX | MAPPO | HAPPO | HASAC |
|---|---|---|---|---|---|---|---|---|
| Hidden layer size | 64 | 64 | 64 | 64 | 64 | 64 | 64 | 64 |
| Learning rate | 1e-4 | 1e-4 | 1e-4 | 1e-4 | 1e-4 | 3e-4 | 3e-4 | 5e-4 |
| Batch size | 32 | 32 | 32 | 32 | 32 | 10 | 10 | 10 |
| Discount | 0.99 | 0.99 | 0.99 | 0.99 | 0.99 | 0.99 | 0.99 | 0.99 |
| Target update interval | 200 | 200 | 200 | 200 | 200 | 200 | 200 | 50 |
| Number of steps to look ahead | 1 | - | - | - | - | 5 | 5 | 20 |
| Entropy coef | - | - | - | - | - | 0.01 | 0.01 | - |
| Grad norm clip | 10 | 10 | 10 | 10 | 10 | 10 | 10 | - |

More detailed information can be found in our code available at `https://github.com/lamda-bbo/seq-madac`.

## C   Additional results

### C.1   Results on the original Sigmoid benchmark

We compare our method with the famous decomposition value-based method VDN [39], as well as the widely used value-based single-agent RL algorithm DQN [28] on the original Sigmoid benchmark. As shown in Figure 4, our method can achieve competitive and even better results compared to the effective decomposition-based VDN when there are no strong inter-dependencies among the parameters. Moreover, the single-agent RL algorithm DQN fails to learn effectively because of the combinatorial explosion of the action space on the 5D Sigmoid, and even fails to train on the 10D Sigmoid, which is also observed in [44], emphasizing the importance of the multi-agent modeling.

### C.2   Comparison of the correct order and the reserve order on Seq-Sigmoid and Seq-Sigmoid-Mask

The comparison of the sequential methods based on the correct order and the reverse order is shown in Figure 5. Basically, the reverse order produces lower performance and larger variance than the correct order due to the wrong capture of the inherent inter-dependencies between the parameters, which misleads the learning of the whole team. There is an exception, HAPPO, which only considers the update order of the agents rather than the order of taking actions. HAPPO avoids suffering from

---

[2]`https://github.com/uoe-agents/epymarl`

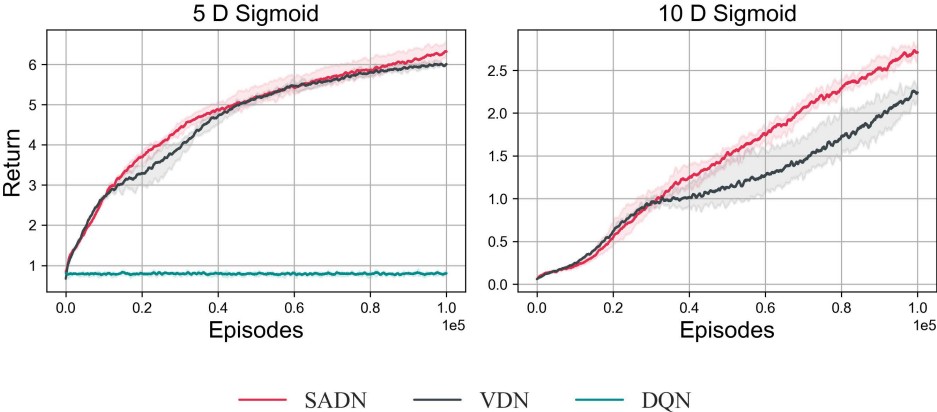

Figure 4: Training curves of the return value obtained by the compared methods on the original Sigmoid benchmark.

the misleading of the reverse order, but fails to exploit the sequential information of the correct order as well. The comparison of using the correct and the reverse order provides evidence for the effectiveness of modeling the inherent inter-dependencies correctly, which also leads to more efficient training.

### C.3   Comparison of the correct order and the reserve order on MOEA/D

The results in Table 4 demonstrate the effectiveness of correctly considering the parameter order on the real-scenario multi-objective evolutionary algorithm. Generally, the correct order produces better results than the reverse order within the same sequential methods, which evidently shows that the sequential information of the correct order can boost better performance, and the improvement of the performance comes from the correct order, rather than simply making the actions selected by the previous agents available to the subsequent agents. There are two exceptions: for MAPPO and SAQL, the rank using the reverse order is better than that using the correct order. This is because these two methods cannot efficiently make full use of the action sequence information, causing the performance of the correct order to occasionally be far from optimal, and sometimes even worse than the reverse order. For MAPPO, it is not originally designed to capture the sequence information, and we basically add the previous agent's action to the next agent's state, according to [11]. For SAQL, as an extension to IQL [40], it lacks reward allocation and fails to reveal the interactions among the agents, which leads to a weak ability to make use of the action sequence information.

### C.4   Discussion on generalization ability with different dimensions

We test the cross-dimension results of SADN on the MOEA/D benchmark as shown in Table 5. Overall, the results demonstrate that our proposed method exhibits good generalization ability across different dimensions. However, we acknowledge that cross-dimensional generalization remains a significant challenge, and we will further investigate and improve its generalization capability in the future, such as through multi-dimensional mixed training and employing multi-head models.

### C.5   Runtime analysis of different methods

We test the training time of the compared methods on the 10D Seq-Sigmoid benchmark for 1,050,000 steps and the MOEA/D benchmark for 405,000 steps. The results are shown in Table 6.

Due to the independent and simultaneous update of the individual advantage nets, SADN demonstrates good efficiency in terms of training and inference time, noting that we test the learned policies at fixed intervals during the training process. In summary, to achieve the best average rank, the running time

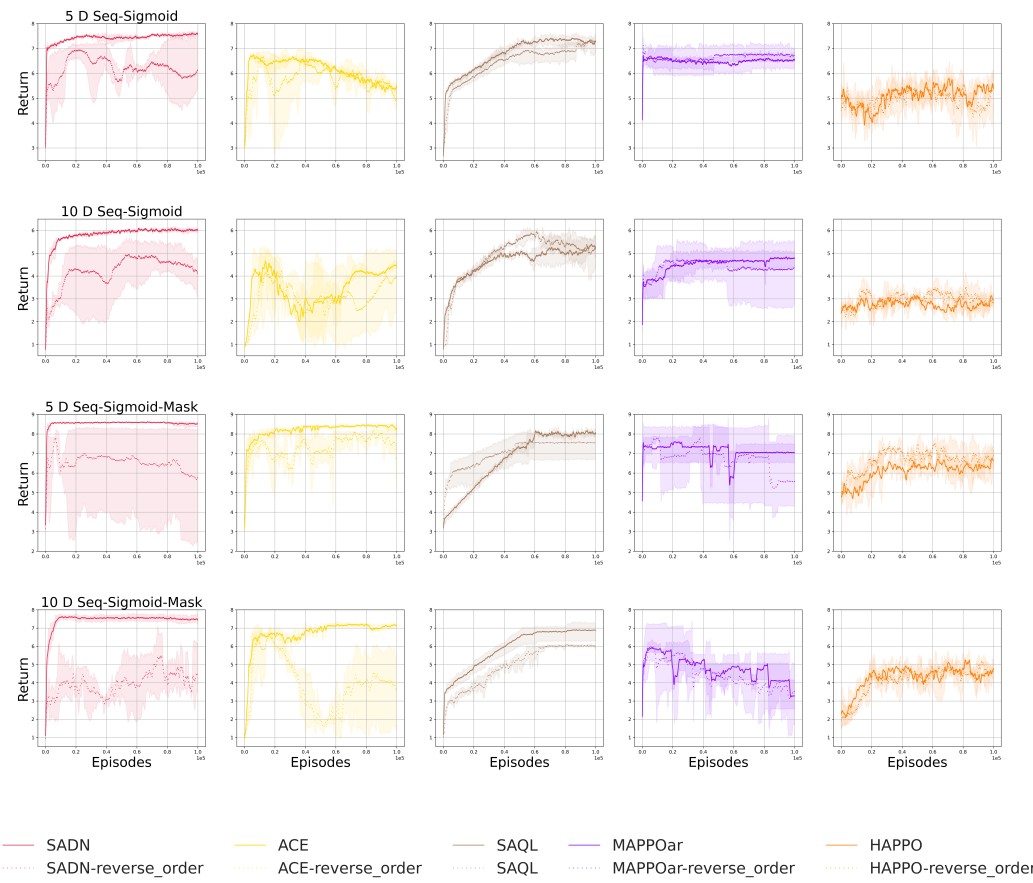

Figure 5: Training curves of return value obtained by correct order and reverse order on four Seq-Sigmoid variant tasks, where the results are averaged over 3 runs.

of our SADN is only slower than two value decomposition-based methods, i.e., VDN and QMIX, while faster than SAQL, ACE, as well as the policy gradient-based MAPPO, HAPPO, and HASAC.

Table 4: IGD values of the compared methods with the correct order and reverse order (denoted as -r) on different problems. Each result consists of the mean of 10 runs. The best mean value on each problem is highlighted in bold. The symbols '+', '−' and '≈' indicate that the result is significantly superior to, inferior to, and almost equivalent to our method SADN, respectively, according to the Wilcoxon rank-sum test with significance level 0.05. The top three problems are the problems for training, and the bottom five problems are for testing.

| Problem | Dim | MOEA/D | SADN | SADN-r | ACE [19] | ACE-r | SAQL [3] | SAQL-r | MAPPOar [11] | MAPPOar-r |
|---------|-----|--------|------|--------|----------|-------|----------|--------|--------------|-----------|
| DTLZ2 | 6 | 4.593e-02 | 3.809e-02 | 3.819e-02 | 3.851e-02 | 4.615e-02 | 3.950e-02 | 4.061e-02 | 3.804e-02 | **3.786e-02** |
|  | 9 | 4.598e-02 | **3.875e-02** | 4.158e-02 | 4.057e-02 | 4.619e-02 | 4.337e-02 | 4.354e-02 | 4.308e-02 | 4.167e-02 |
|  | 12 | 4.599e-02 | **3.874e-02** | 4.176e-02 | 4.063e-02 | 4.634e-02 | 4.698e-02 | 4.734e-02 | 3.886e-02 | 4.284e-02 |
| WFG4 | 6 | 5.729e-02 | **4.537e-02** | 4.847e-02 | 4.626e-02 | 6.282e-02 | 4.817e-02 | 4.747e-02 | 4.677e-02 | 4.576e-02 |
|  | 9 | 5.736e-02 | **5.190e-02** | 6.026e-02 | 5.853e-02 | 6.725e-02 | 5.365e-02 | 5.375e-02 | 5.471e-02 | 5.356e-02 |
|  | 12 | 5.751e-02 | **5.213e-02** | 6.021e-02 | 5.751e-02 | 6.983e-02 | 6.241e-02 | 5.958e-02 | 5.351e-02 | 5.797e-02 |
| WFG6 | 6 | 5.855e-02 | **3.908e-02** | 4.027e-02 | 3.962e-02 | 5.811e-02 | 3.964e-02 | 3.995e-02 | 3.944e-02 | 3.921e-02 |
|  | 9 | 6.641e-02 | 4.884e-02 | 7.757e-02 | 5.111e-02 | 5.992e-02 | **4.384e-02** | 4.432e-02 | 6.093e-02 | 5.949e-02 |
|  | 12 | 6.864e-02 | **4.837e-02** | 7.674e-02 | 5.490e-02 | 7.908e-02 | 5.676e-02 | 5.755e-02 | 5.786e-02 | 6.710e-02 |
| Training: +/−/≈ |  | 0/9/0 |  | 0/8/1 | 0/8/1 | 0/9/0 | 1/8/0 | 1/8/0 | 0/7/2 | 1/8/0 |
| DTLZ4 | 6 | 5.455e-02 | **3.895e-02** | 3.980e-02 | 4.221e-02 | 5.598e-02 | 5.019e-02 | 4.856e-02 | 3.932e-02 | 4.008e-02 |
|  | 9 | 6.229e-02 | 7.319e-02 | 5.366e-02 | **4.492e-02** | 5.765e-02 | 6.895e-02 | 5.833e-02 | 6.252e-02 | 5.847e-02 |
|  | 12 | 6.790e-02 | 7.852e-02 | 5.196e-02 | **4.650e-02** | 6.667e-02 | 8.589e-02 | 8.688e-02 | 5.336e-02 | 6.211e-02 |
| WFG5 | 6 | 6.303e-02 | 4.749e-02 | 4.748e-02 | 4.752e-02 | 6.145e-02 | 5.390e-02 | 5.539e-02 | **4.713e-02** | 4.719e-02 |
|  | 9 | 6.351e-02 | **4.773e-02** | 4.778e-02 | 4.792e-02 | 6.156e-02 | 4.788e-02 | 4.787e-02 | 5.070e-02 | 4.798e-02 |
|  | 12 | 6.381e-02 | 4.793e-02 | 4.759e-02 | 4.791e-02 | 6.169e-02 | 4.954e-02 | 4.983e-02 | **4.751e-02** | 4.980e-02 |
| WFG7 | 6 | 5.803e-02 | **3.893e-02** | 3.994e-02 | 3.942e-02 | 5.819e-02 | 3.958e-02 | 3.980e-02 | 3.922e-02 | 3.907e-02 |
|  | 9 | 5.812e-02 | **4.053e-02** | 4.954e-02 | 4.470e-02 | 5.878e-02 | 4.257e-02 | 4.235e-02 | 4.285e-02 | 4.154e-02 |
|  | 12 | 5.814e-02 | **4.060e-02** | 4.950e-02 | 4.447e-02 | 5.936e-02 | 4.975e-02 | 4.986e-02 | 4.115e-02 | 4.437e-02 |
| WFG8 | 6 | **7.875e-02** | 1.018e-01 | 1.077e-01 | 1.029e-01 | 1.177e-01 | 1.054e-01 | 1.031e-01 | 1.005e-01 | 1.026e-01 |
|  | 9 | 9.680e-02 | **7.853e-02** | 9.191e-02 | 9.787e-02 | 1.053e-01 | 9.716e-02 | 9.299e-02 | 7.947e-02 | 7.911e-02 |
|  | 12 | 8.710e-02 | 7.891e-02 | 9.195e-02 | 8.387e-02 | 9.330e-02 | 9.023e-02 | 8.947e-02 | **7.873e-02** | 8.164e-02 |
| WFG9 | 6 | 5.600e-02 | **3.987e-02** | 4.053e-02 | 4.012e-02 | 5.495e-02 | 4.010e-02 | 4.219e-02 | 3.994e-02 | 3.975e-02 |
|  | 9 | 5.748e-02 | 4.341e-02 | 5.831e-02 | 4.552e-02 | 5.562e-02 | 4.281e-02 | **4.168e-02** | 4.677e-02 | 4.446e-02 |
|  | 12 | 5.827e-02 | **4.249e-02** | 5.839e-02 | 4.477e-02 | 7.265e-02 | 7.291e-02 | 6.463e-02 | 5.299e-02 | 4.986e-02 |
| Testing: +/−/≈ |  | 1/12/2 |  | 3/10/2 | 2/9/4 | 1/13/1 | 0/11/4 | 2/11/2 | 3/8/4 | 4/9/2 |
| average rank | 6 | 7.5 | **2** | 5.625 | 4.5 | 8.625 | 5.75 | 6.25 | 2.5 | 2.25 |
|  | 9 | 7.375 | **2.5** | 5.5 | 4.75 | 7.5 | 4.375 | 3.625 | 5.625 | 3.75 |
|  | 12 | 6.25 | **2.25** | 5.25 | 2.75 | 8 | 6.75 | 6.875 | 2.5 | 4.375 |

Table 5: Cross-dimension test of SADN on the MOEA/D benchmark. SADN-n refers to the model trained exclusively on problems with n dimensions.

|  | dim | MOEA/D | SADN-6 | SADN-9 | SADN-12 |
|---|---|---|---|---|---|
| DTLZ2 | 6 | 4.593 | **3.809** | 3.854 | 3.810 |
|  | 9 | 4.598 | 3.928 | 3.875 | **3.840** |
|  | 12 | 4.599 | 3.936 | 3.882 | **3.874** |
| WFG4 | 6 | 5.729 | 4.537 | 4.543 | **4.521** |
|  | 9 | 5.736 | 4.973 | 5.190 | **4.922** |
|  | 12 | 5.751 | 5.260 | 5.322 | **5.213** |
| WFG6 | 6 | 5.855 | **3.908** | 3.919 | 3.913 |
|  | 9 | 6.641 | 5.044 | **4.884** | 5.010 |
|  | 12 | 6.864 | 5.036 | 5.138 | **4.837** |
| DTLZ4 | 6 | 5.455 | **3.895** | 3.970 | 3.968 |
|  | 9 | 6.229 | 5.700 | 7.319 | **5.587** |
|  | 12 | **6.790** | 7.593 | 8.176 | 7.852 |
| WFG5 | 6 | 6.303 | **4.749** | 4.764 | 4.759 |
|  | 9 | 6.351 | 4.814 | **4.773** | 4.788 |
|  | 12 | 6.381 | 4.824 | 4.823 | **4.793** |
| WFG7 | 6 | 5.803 | 3.893 | **3.886** | 3.893 |
|  | 9 | 5.812 | **3.980** | 4.053 | 3.990 |
|  | 12 | 5.814 | 4.176 | 4.159 | **4.060** |
| WFG8 | 6 | **7.875** | 10.18 | 10.56 | 10.67 |
|  | 9 | 9.680 | 9.020 | **7.853** | 8.948 |
|  | 12 | 8.710 | 8.013 | 8.191 | **7.891** |
| WFG9 | 6 | 5.600 | **3.987** | 3.997 | 3.989 |
|  | 9 | 5.748 | 4.387 | **4.341** | 4.370 |
|  | 12 | 5.872 | 4.274 | 4.423 | **4.249** |
| average rank |  | 3.708 | 2.083 | 2.5 | **1.708** |

Table 6: The training time of the compared methods on the 10D Seq-Sigmoid benchmark for 1,050,000 steps and the MOEA/D benchmark for 405,000 steps

|  | SADN | ACE | SAQL | VDN | QMIX | MAPPO | HAPPO | HASAC |
|---|---|---|---|---|---|---|---|---|
| 10DSeq-Sigmoid | 4h28' | 5h01' | 6h34' | 3h23' | 3h29' | 12h32' | 16h39' | 21h17' |
| MOEA/D | 14h56' | 16h20' | 19h46' | 14h48' | 15h23' | 17h44' | 20h02' | 23h52' |

