# OpenReview forum: "Sequential Multi-Agent Dynamic Algorithm Configuration"
_NeurIPS.cc/2025/Conference — NeurIPS 2025 poster_

### Official Review · Reviewer_xzDh · 2025-06-29

**Clarity:** 3
**Significance:** 3
**Originality:** 3
**Rating:** 5
**Confidence:** 5

**Summary:**

The work proposes an improvement for multi-agent dynamic algorithm configuration by making hyperparameter interdependence explicit to the agent. The work evaluates the effectiveness of this improvement on a variety of whitebox benchmarks, with controllable interdependence as well as on real optimization scenarios from the dynamic algorithm configuration community. This evaluation highlights that explicitly taking hyperparameter interdependence into account can significantly improve algorithm performance and lead to faster learning of successful configuration policies.

**Questions:**

* How does the proposed approach handle continuous hyperparameters?
* How difficult do you think it will be for users to apply your approach to novel DAC problems?
* How large is the action space per agent in the considered scenarios?

**Ethical Concerns:**

["NO or VERY MINOR ethics concerns only"]

**Final Justification:**

The work proposes an interesting approach to dynamic algorithm configuration. The authors answered all my questions and, with minor revisions to incorporate the overall reviewers feedback, I believe that the work can provide interesting discussions at NeurIPS.

**Limitations:**

The impact of the optimization order is not discussed well in the main paper (see also my comments in the strengths/weaknesses section)

**Quality:**

4

**Strengths And Weaknesses:**

I very much like the work and particularly appreciate that whitebox benchmarks are used to evaluate a variety of challenges that could be faced in real world optimization problems. Further, I commend that the experiment evaluating the influence of action ordering is also performed on the real world scenario and not only on white-box benchmarks. Thus, I overall am very positive about this work and the empirical evaluation.

The biggest weakness in my opinion is the brevity of the discussion about the influence of hyperparameter ordering in the setup. While the appendix provides additional insights about the influence of the ordering, it is barely discussed in the main paper. The multi-agent DAC setting, as presented in this paper, requires quite some prior knowledge of a user to properly setup the configuration space and overall scenario. This is a significant limitation for the proposed method to be practical to real-world use cases and thus should be addressed in more detail in the main paper and not just in the appendix.

The description of the white-box benchmarks in the appendix shows that there is a parameter C that determines the action space size of an agent. I could not find how this parameter was set in the presented experiments.

I believe there +/-/approx column in the Tables might not be correct or at least difficult to read. For example in Table 4 SAQL/QMIX have 0/2 marked in the + part but 2/1 values highlighted in bold face, indicating the best performance. Either the counts are off or the elements in the table should be highlighted (e.g. italicized if they are in the + or underlined if they are in the approx category) differently to make it easier to match the results.

---

> ### Author Rebuttal · Authors · 2025-07-31
>
> Thanks for your valuable and encouraging comments! Below please find our response.
>
> ## Q1 The brevity of the discussion about the influence of hyperparameter ordering in the setup.
>
> Thank you for providing such constructive insights. In practice, it is indeed a significant problem to determine a proper order of the hyperparameters, and here we provide a simple and natural way to determine a right order, that is, setting the order in which the hyperparameters take effect during the execution of the algorithm. Since we will know when certain hyperparameters take effect in the target algorithm if we want to dynamically configure it, this order can be relatively easy to obtain and treated as an order with acceptable performance. However, we have to point out that such an order may not always be optimal with respect to training efficiency and effectiveness. We believe that determining the best order for the hyperparameters of general algorithms can be left as challenging and promising future works, which may include applying large language models to leverage their knowledge to set the proper order, as well as causal models to automatically learn the hyperparameters' causal structure from the data and further determine the proper order using the topological order of the learned causal graph.
>
> ## Q2 How the parameter C is set?
>
> We are sorry for not making it clear. In the Sigmoid-series benchmarks, we have the agents' action space defined as follows:
> $$
> a_{h,t} \in \left \\{ \frac{0}{C_h}, \frac{1}{C_h}, \dots, \frac{C_h-1}{C_h} \right\\}.
> $$
> We set $C_h$ as 10, which is a relatively large action space for the agents to choose actions from. We will revise to make it clear. Thank you for pointing it out.
>
> ## Q3 Some concerns about the table notations.
>
> Thank you for the valuable suggestions. However, there are some misunderstandings that need to be clarified.
>
> We separately summarize the Wilcoxon rank-sum test results in the training problem instances and the testing problem instances. One result highlighted in bold denotes that it is the best result, but not necessarily significantly superior to our method SADN (marked with "+"), according to the Wilcoxon rank-sum test.
>
> For example, as you mentioned (actually in Table 1, we believe), SAQL has two bold results, one in the training part and the other in the testing part. In the testing part, SAQL is marked as 0/11/4 because its only bold result in the testing part is almost equivalent to our method SADN according to the Wilcoxon rank-sum test. While QMIX is marked as 2/11/2 in the testing part, because it has two results significantly superior to our method SADN (i.e., DTLZ4 with dim 9 and 12). These two results of QMIX are not highlighted in bold because they are not the best (VDN achieves the best results on DTLZ4 with dim 9 and 12).
>
> We gladly take the advice of highlighting them differently, e.g., make the values in the table italic or underlined if they are in different categories of significant test. We will revise it accordingly to make the tables easier to read. Thank you very much.
>
> ## Q4 How does the proposed approach handle continuous hyperparameters?
>
> Thank you for the question. In the MOEA/D environment, we have 4 agents configuring 4 types of hyperparameters:
> 1. Agent 1 is responsible for adjusting the neighborhood size, which controls the maximum distance between solutions and their neighbors. Agent 1 can choose the neighborhood size from the action space {15, 20, 25, 30};
>
> 2. Agent 2 is responsible for determining the types of reproduction operators with action space {0, 1, 2, 3}, denoting the different operator types shown as follows;
>
> 3. Agent 3 is responsible for determining the parameters of reproduction operator, scaling factor F specifically, with action space {0.4, 0.5, 0.6, 0.7};
>
> 4. Agent 4 is responsible for determining whether to update the weights to bring in new subproblems, with action space {0, 1}, 1 for updating and 0 for not updating the weights.
>
> The detailed information about the operators is shown below:
> - OP1:
>     $$\boldsymbol {x}'^{(i)}=\boldsymbol{x}^{(i)}+F \times\left(\boldsymbol{x}^{(r_{1})}-\boldsymbol{x}^{(r_{2})}\right),$$
>
> - OP2:
>     $$
>     \boldsymbol {x}'^{(i)}=\boldsymbol{x}^{(i)}+F \times\left(\boldsymbol{x}^{(r_{1})}-\boldsymbol{x}^{(r_{2})}\right)+F \times\left(\boldsymbol{x}^{(r_{3})}-\boldsymbol{x}^{(r_{4})}\right),
>     $$
> - OP3:
>     $$
>     \boldsymbol {x}'^{(i)}=\boldsymbol{x}^{(i)}+K \times\left(\boldsymbol{x}^{(i)}-\boldsymbol{x}^{(r_{1})}\right)+F \times\left(\boldsymbol{x}^{(r_{2})}-\boldsymbol{x}^{(r_{3})}\right)+F \times\left(\boldsymbol{x}^{(r_{4})}-\boldsymbol{x}^{(r_{5})}\right),
>     $$
> - OP4:
>     $$
>     \boldsymbol {x}'^{(i)}=\boldsymbol{x}^{(i)}+K \times\left(\boldsymbol{x}^{(i)}-\boldsymbol{x}^{(r_{1})}\right)+F \times\left(\boldsymbol{x}^{(r_{2})}-\boldsymbol{x}^{(r_{3})}\right).
>     $$
>
> We handle continuous hyperparameters by discretizing the continuous action space into a discrete action space, which is a common practice when dealing with the continuous action space. Empirically, discretization has achieved good performance. In the future works, we could apply other techniques to directly handle continuous action, e.g., determine the value by the probability distribution output of the actor network.
>
> ## Q5 How difficult do you think it will be for users to apply your approach to novel DAC problems?
>
> Thank you for the constructive question. We believe it is relatively easy to apply our approach to real-scenario DAC problems.
>
> Firstly, the users need to follow some common procedures to address the DAC problems:
>
> - Construct their target algorithm as an environment that receives multiple hyperparameters and executes step by step.
>
> - Set the configuration space of the multiple hyperparameters in their target algorithm.
>
> Secondly, the users need to determine the hyperparameter order. They can either use our recommended order described in Q1 or set the order according to their own expert knowledge.
>
> Moreover, our method has a relatively smaller amount of hyperparameters compared to policy gradient-based MARL algorithms. Empirically, we find that our method does not require delicate hyperparameter tuning to achieve good performance. The default setting in our code (config/alg/sadn_ns.yaml) can boost good performance in many cases. You can also conveniently and simply modify it based on your experience.
>
> In conclusion, we believe that our method is user-friendly and has the potential to be widely applied to solve the DAC problems.
>
> ## Q6 How large is the action space per agent in the considered scenarios?
>
> Thank you for the question. In the white-box benchmarks (the Seq-Sigmoid series), the dimension of the action space per agent is 10. In the MOEA/D, the dimension of the action space for Agents 1 to 3 is 4, and the dimension of the action space for Agent 4 is 2.
>
> ---
>
> **We hope that our response has addressed your concerns, but if we missed anything, please let us know.**

---

> > ### Comment · Reviewer_xzDh · 2025-08-01
> >
> > Thank you very much for the clarifications. I  have read all reviews and responses and remain confident in my assessment that this paper should be accepted.

---

> > > ### Author Response · Authors · 2025-08-05
> > >
> > > Thank you for your encouraging feedback and positive rating! We sincerely appreciate the time and effort you have dedicated to reviewing our paper and providing thoughtful comments. Thank you.

---

### Official Review · Reviewer_aCAk · 2025-07-02

**Clarity:** 3
**Significance:** 2
**Originality:** 2
**Rating:** 4
**Confidence:** 3

**Summary:**

The paper presents a new method to deal with inherent inter-dependencies between parameters in Dynamic Algorithm Configuration (DAC). First, the authors propose a sequential multi-agent formulation for the DAC problem, where each agent acts based on what all previous agents’ decisions at the same timestep. Then, they propose a sequential advantage decomposition network, where each agent models its own advantage function and show that sequentially selecting the actions that maximize the individual agents’ advantages is equivalent to selecting the best global action. The individual advantages are then combined to find the global advantage, which is used together with a learned global value function to jointly train all the agents. The presented framework is tested on several white-box environments, as well as several real-scenarios DAC benchmarks, showing very good performances against state-of-the-art MARL and sequential methods used in DAC, especially when the number of dimensions (configurable parameters) increases.

**Questions:**

1. In 4.2 you say that during training the instance is resampled every time an episode ends. Is the set of testing instance still disjunct from the training one?

2. As I understand, every parameter to configure corresponds to a different agent. In the MOEA/D environments, do all the operators have the same number of parameters?
  If so, how is this handled (e.g. are agents correspondingly enabled and disabled or are their actions ignored?);
  If not, is this not a limitation, as it can often happen that the number of subsequent decision depend on the previous ones?

3. Can you please discuss the performance of the different methods in terms of training and evaluation time?

**Ethical Concerns:**

["NO or VERY MINOR ethics concerns only"]

**Final Justification:**

The rebuttal has clarified the questions I had.

**Limitations:**

yes

**Quality:**

3

**Strengths And Weaknesses:**

Strengths:

•	The paper is well written and easy to follow.

•	The empirical experiments cover a fair number of white-box and more realistic and complex environments and show convincing performances against several state-of-the-art approaches.

•	While the selected application domain allows the authors to clearly explain the problem of decision inter-dependency, the proposed method can be easily applied to many other multi-agent problems.

Weaknesses:

•	The proposed method is mostly a combination of multi-agent advantage decomposition and value decomposition network.

•	It is not very clear how the proposed Seq-MADAC differs from the SEMDP, besides from the addition of the contextual component.

•	It is not clear whether the method can deal with environments where the number of parameters to configure at each timestep changes depends on decisions taken early in the sequence.

---

> ### Author Rebuttal · Authors · 2025-07-31
>
> Thanks for your valuable and encouraging comments! Below please find our response.
>
>
> ## Q1 The difference between the proposed Seq-MADAC and the SEMDP, besides from the addition of the contextual component.
>
> Thank you for the question. We formulate the Seq-MADAC task as a contextual sequential MMDP. Besides from the addition of the contextual component, the main difference between the proposed contextual sequential MMDP and the SE-MDP is that we formulate this task as a multi-agent MDP, while the SE-MDP is a single-agent MDP. We have given the definition of MMDP in line 114. Moreover, we allow communication between the agents. The agents basically execute in a decentralized manner, with only the currently taken action being mutually notified.
>
> ## Q2 Can the method deal with environments with the changing number of parameters?
>
> Thank you for the insightful and constructive question! Although we did not address this issue in our paper, we believe it can be left as a promising future work just as discussed in the Q4 of Reviewer FPTa. ConfigX [1] proposes a novel modularization system that supports flexible reconstruction of different algorithm sub-modules. Inspired by the exploration of modular system in ConfigX, we can modularize the algorithm, and when we choose different modules, the corresponding number of parameters varies. We can have one agent responsible for the parameter configuration of one module, with one leader agent to schedule the sub-modules. Moreover, this modeling also exhibits obvious inter-dependencies in the action sequence (e.g., module type and module configuration), where our method may achieve good performance. Besides, we may enable or disable some agents to deal with the changing number of parameters. For example, when extra parameters are activated due to the previous configurations, the corresponding agents are enabled. We will add this discussion into our revised paper. Thank you very much.
>
> [1] ConfigX: Modular Configuration for Evolutionary Algorithms via Multitask Reinforcement Learning. AAAI, 2024.
>
> ## Q3 Is the set of testing instance still disjunct from the training one?
>
> Yes, we resample the testing instance from the problem instance distribution, once one episode is done and the environment is reset.
>
> ## Q4 In the MOEA/D environments, do all the operators have the same number of parameters?
>
> In the MOEA/D environment, we have 4 agents configuring 4 types of hyperparameters:
> 1. Agent 1: neighborhood size, which controls the maximum distance between solutions and their neighbors;
>
> 2. Agent 2: types of reproduction operators shown as follows;
>
> 3. Agent 3: parameters of reproduction operator, the scaling factor F in specific;
>
> 4. Agent 4: whether to update the weights to bring in new subproblems.
>
> The operators have a different number of parameters, which are shown below:
>
> - OP1:
>     $$\boldsymbol {x}'^{(i)}=\boldsymbol{x}^{(i)}+F \times\left(\boldsymbol{x}^{(r_{1})}-\boldsymbol{x}^{(r_{2})}\right),$$
>
> - OP2:
>     $$
>     \boldsymbol {x}'^{(i)}=\boldsymbol{x}^{(i)}+F \times\left(\boldsymbol{x}^{(r_{1})}-\boldsymbol{x}^{(r_{2})}\right)+F \times\left(\boldsymbol{x}^{(r_{3})}-\boldsymbol{x}^{(r_{4})}\right),
>     $$
> - OP3:
>     $$
>     \boldsymbol {x}'^{(i)}=\boldsymbol{x}^{(i)}+K \times\left(\boldsymbol{x}^{(i)}-\boldsymbol{x}^{(r_{1})}\right)+F \times\left(\boldsymbol{x}^{(r_{2})}-\boldsymbol{x}^{(r_{3})}\right)+F \times\left(\boldsymbol{x}^{(r_{4})}-\boldsymbol{x}^{(r_{5})}\right),
>     $$
> - OP4:
>     $$
>     \boldsymbol {x}'^{(i)}=\boldsymbol{x}^{(i)}+K \times\left(\boldsymbol{x}^{(i)}-\boldsymbol{x}^{(r_{1})}\right)+F \times\left(\boldsymbol{x}^{(r_{2})}-\boldsymbol{x}^{(r_{3})}\right).
>     $$
>
> OP1 and OP2 have only one operator parameter, which is $F4$, while OP3 and OP4 have two operator parameters, namely $K$ and $F$. In our setting, agent 3 configures the operator $F$, a common parameter shared by all operators, with an action space of {0.4, 0.5, 0.6, 0.7}, while keeping factor $K$ fixed at 0.5. More details can be found in Appendix B.2. We will revise the text to make this clearer. Thank you.
>
> ## Q5 Runtime analysis of different methods.
>
> Thank you for the valuable question. We agree that adding runtime analysis will further strengthen our results. We tested the training time of the compared methods on the 10D Seq-Sigmoid benchmark for 1,050,000 steps and the MOED/D benchmark for 405,000 steps. The results are shown below:
>
> | |SADN|ACE|SAQL|VDN|QMIX|MAPPO|HAPPO|HASAC|
> |---|---|---|---|---|---|---|---|---|
> |10DSeq-Sigmoid|4h28'|5h01'|6h34'|3h23'|3h29'|12h32'|16h39'|21h17'|
> MOEA/D|14h56'|16h20'|19h46'|14h48'|15h23'|17h44'|20h02'|23h52'|
>
> Due to the independent and simultaneous update of the individual advantage nets, SADN demonstrates good efficiency in terms of inference time. In summary, to achieve the best average rank, the running time of our SADN is only slower than two value decomposition-based methods, i.e., VDN and QMIX, while faster than SAQL, ACE, as well as the policy gradient-based MAPPO, HAPPO, and HASAC. We will include these results into our main paper. Thank you very much!
>
> ---
>
> **We hope that our response has addressed your concerns, but if we missed anything, please let us know.**

---

> ### Comment · Reviewer_aCAk · 2025-08-05
>
> Thanks for your explanations and additional results, which answered my questions.
> I think most of my doubts came from the fact that you use “agents” when discussing ACE (lines 152-163), which makes it seem like that approach is also multi-agent.
> I think that should be fixed so that the ACE algorithm is described correctly, to avoid misunderstandings.

---

> > ### Author Response · Authors · 2025-08-06
> >
> > Thanks for your feedback! We are glad to hear that your concerns have been addressed. We will provide a more explicit description of the ACE algorithm in our revised version, according to your valuable advice. Thank you.

---

### Official Review · Reviewer_FPTa · 2025-07-03

**Clarity:** 3
**Significance:** 3
**Originality:** 3
**Rating:** 5
**Confidence:** 4

**Summary:**

This paper introduces an improved version of multi-agent DAC framework, where the authors make in-depth exploration on the sequential relationship among the hyper-parameters within a relatively complex algorithm configuration space. To this end, the authors decompose the per-step parameter control as sequential steps, where each step is made by a separate RL agent considering both the state information and the decisions of previous agents. The authors give a basic proof of effectiveness of such decomposition modelling and apply this scheme to multi-agent RL framework, which is proposed by the authors and termed as SDAN (generally a VDN structure). The effectiveness of SDAN is validated on multiple DAC tasks including a complex one: configuring MOEA/D, a state-of-the-are multi-objective optimization algorithm. The results demonstrate that SDAN can be generally utilized for different DAC tasks, its learning effectiveness is compared with many MARL baselines.

**Questions:**

See Strengths And Weaknesses.

**Ethical Concerns:**

["NO or VERY MINOR ethics concerns only"]

**Final Justification:**

This paper presents an interesting idea supported by promising experimental results. The authors have also effectively addressed my initial concerns in the rebuttal. I recommend acceptance.

**Limitations:**

yes

**Quality:**

3

**Strengths And Weaknesses:**

Strengths:

1. This paper provides a new perspective to rethink the MDP modelling for automated algorithm design, especially, a sub-task: algorithm configuration. Given the learning for algorithm design gets popular in recent ML researches, the significance of this paper is for sure.
2. The motivation of this paper is clear and reasonable, since for complex algorithm design task, the submodule in the algorithms, together with the related hyper-parameter values, are somewhat correlated with with each other, hence modelling them with inter-relationship is very crucial to learn some effective configuration policies.
3. The experiment is solid. I especially appreciate the authors to compare with multiple MARL baselines.

Weaknesses:

1. The writing and presentation of this paper has a large room to improve. For example, there are too many symbols and equations and lengthy elaboration of MARL, including both the related works and methodology, which can be easily organized into appendix and put more important content into the main body to enhance the motivation, the core design and the philosophy of this paper. Besides, the figures, especially the figure 1, is in poor quality (instance set part). Re-draw this figure to make it more clear the correspondence between the MDP and the algorithm solving instance.
2. For the background (related works), I suggest the authors to enrich the Section 2.1, to include more recent works. For example, a very important and similar work, QMamba (https://openreview.net/forum?id=7qCkf5OT3x), which also decomposes the configuration space to address the inter-relationship of operators and parameter values. This paper should be at least cited and discussed. For Section 2.2, I suggest move some MARL basic knowledge to appendix, shorten this section and leave the most important part, that is, what is MARL and the limitation of naive MARL on complex DAC tasks. Besides, there are many sequential MDP approaches while this paper reviews few of them (such as Qtransformer , SDQN, etc).
3. I think the most confusing part in the decomposition scheme is the order of the actions. How to ensure the proposed SADN is robust to the different orders of hyper-parameters within the same algorithm? I can not find any special design to address this, only an experiment in appendix to show the effectiveness of correct and reverse order. Besides, the second confusing part is the scalability of SADN, since different algorithms may hold various number of hyper-parameters, I guess SADN can not be trained simultaneously on diverse algorithm structures and zero-shot to unseen algorithms. This is somehow inefficient for realistic world. A very promising way to resolve this, I think, is based on your references [10]. this work configx provides a modular algorithm space, which you can use multi-agent system to learn and control.
4. Please include error bars into Table 1.
5. As the learn to optimize domain develops, a more important thing is to analysis why a learning-based approach generalize well on some cases and not well on the others. For example, in Table 1, I observe that SADN underperforms baselines in problem set WFG5, why is that? Providing an analysis on this would significantly help the future works, even though it can not be addressed now.
6. The citation format is wrong, correct it please.

---

> ### Author Rebuttal · Authors · 2025-07-31
>
> Thanks for your valuable and encouraging comments! Below please find our response.
>
> ## Q1 and Q2 Suggestions for the writing and presentation improvement.
>
> Thank you very much for your valuable suggestions. We will carefully revise the paper according to your suggestions.
> - We will move the background information of MARL to the appendix and put core theorem proofs into the main body of the paper.
> - We will also re-draw the figures you mentioned to make a clearer demonstration of the detailed information of our method (e.g., the MDP process and the problem instance).
> - We will enrich our related works with more recent and constructive works such as Q-Mamba [1] and Q-Transformer [2]. In the future works, we expect to incorporate more advanced architecture like these works into our algorithm, which may boost a stronger ability to capture action sequence information.
>
> [1] Meta-Black-Box-Optimization through Offline Q-function Learning. ICML, 2025.
>
> [2] Q-Transformer: Scalable Offline Reinforcement Learning via Autoregressive Q-Functions. CoRL, 2023.
>
> ## Q3 How to ensure the proposed SADN is robust to the different orders of hyper-parameters within the same algorithm? Setting of the order?
>
> Thank you for your insightful comment! In our setting, the number and order of hyperparameters must be determined before training and remain fixed during the algorithm's execution. Therefore, the setting you mentioned regarding different orders of hyperparameters within the same algorithm is beyond the scope of our framework. However, we fully acknowledge the importance of this setting, as it represents the next step in the DAC field and can be regarded as multi-algorithm configuration or universal algorithm configuration. Leveraging the ConfigX framework you mentioned may help achieve this, and we will address this discussion specifically in our response to Q4.
>
> As for the setting of hyperparameter order, there is an intuitive and natural way to determine it: setting it according to the sequence in which the hyperparameters take effect during the execution of the algorithm. Specifically, in the code implementation of the target algorithm, all hyperparameters take effect in a specific order, which is the order we have set. For example, in the implementation of MOEA/D, after initializing the weights, we first determine the neighborhood subproblems (agent 1: neighborhood size), then optimize the subproblems with certain operators (agent 2: types of reproduction operators and agent 3: parameters of reproduction operators), and finally decide whether to adjust the weights (agent 4: whether to update weights) based on the results of this generation. Therefore, it is relatively easy for us to get an order with acceptably good performance.
>
> We will revise to make it clearer. Thank you again for your valuable comment!
>
>
> ## Q4 Concerns about scalability and zero-shot ability of SADN.
>
> Thank you very much for your constructive and insightful question! We believe that modularization is a promising trend towards universal algorithm configuration, and the exploration of modular system in ConfigX [3] is inspiring. ConfigX exhibits obvious inter-dependencies in the action sequence (e.g., module type and module configuration). We believe it is interesting to incorporate our SADN into the ConfigX framework, using the multi-agent modeling to learn and control the module generation and module configuration, which may boost a more universal and general automated algorithm design and configuration. The Q2 of Reviewer aCAk holds similar ideas: the number of parameter changes when we choose different modules. We believe it is a promising future work and will add this discussion to our revised paper. Thank you very much!
>
> [3] ConfigX: Modular Configuration for Evolutionary Algorithms via Multitask Reinforcement Learning. AAAI, 2025.
>
> ## Q5 Suggestions on including error bars into Table 1 and correcting the citation format.
>
> Thank you for your comments. We will carefully revise the paper according to your suggestions.
>
> ## Q6 The issue of SADN underperforming the baseline in one case.
>
> Thank you for the question. As shown in Table 1, there are occasional cases where SADN underperforms the baseline, and all of them occur in the testing phase. This may be owing to the very different function landscape that was not included in the training phase, and the learned RL policy fails to generalize. This issue is always concerned regarding the generalization ability of RL, and we believe good state and reward designs that capture commonalities among various problem sets can further enhance generalization, as well as more diverse training datasets. In general, our SADN shows strong generalization abilities across problem classes.
>
> ---
>
> **We hope that our response has addressed your concerns, but if we missed anything, please let us know.**

---

### Official Review · Reviewer_HrJN · 2025-07-03

**Clarity:** 3
**Significance:** 3
**Originality:** 4
**Rating:** 5
**Confidence:** 4

**Summary:**

This paper considers the problem of dynamically configuring multiple hyperparameters of an algorithm, where the hyperparameters have a sequential order to tune. The authors proposed a contextual sequential MMDP formulation and the corresponding method, the sequential advantage decomposition network. Extensive experiments on some artificial benchmarks and the real-world task of tuning hyperparameters of the popular multi-objective evolutionary algorithm MOEA/D show the effectiveness of the proposed method.

**Questions:**

- Why can the proposed SADN avoid the long sequence update issue of ACE?
- How to determine the right order of hyperparameters in practice?
- For Seq-Sigmoid, it’s not clear to me why the change of the formulation (as shown in line 249) can avoid the influence on the former hyperparameter.
- line 254, What do you mean by $\mathcal{N}(T/2,T/4,H)$?
- line 255, clearly mention $n < H$?
- Though introduced in the appendix, I think the sequential order of the four hyperparameters of MOEA/D should be introduced in the main paper, which can ease the understanding of the readers.
- Figures 2 & 3, are three runs sufficient?
- Table 1, are training and testing on different benchmarks with the same number of objectives? If so, how about the performance with a different number of objectives, e.g., training on 6 objectives while testing on 9 objectives? Such a test can further examine the generalization ability.
- I can understand the IGD metric used in the experiments of MOEA/D, but I suggest that a brief introduction is necessary, as the readers may not be familiar with multi-objective optimization.
- line 890-891, how to calculate the sparsity of each solution when the size of the elite population exceeds the capacity?
- line 937-941, you used the default hyperparameter settings for VDN, QMNIX, and MAPPO. Then, how to make them the same for fair comparison?
- Table 4, for SAQL and MAPPO with 9 objectives, the rank using the reverse order is better than that using the correct order. This seems to be strange. Can you give some explanation?

**Ethical Concerns:**

["NO or VERY MINOR ethics concerns only"]

**Final Justification:**

Thanks for the detailed response. My questions have been addressed well. I am happy to increase my score by 1.

**Limitations:**

Yes, the authors mentioned the difficulty of determining the right order of adjusting the hyperparameters in the conclusion part.

**Paper Formatting Concerns:**

The paper is generally well written. Some minor problems:
- Line 75, “by leverage” -> “by leveraging”
- Line 99, “which can also” -> “which may”
- Line 211, $\pi$ -> $\bm{\pi}$  The mathematical symbols should be consistent throughout the paper.
- Line 246, $\alpha_{h-1}$ -> $\alpha_h$. The same issue at line 832.
- Line 252, the space before $)$ is redundant.
- Line 293, “experiment results” -> “experimental results”
- Table 1, “2.” The same issue in Table 4.
- Line 794, the period is missing. The same issue at line 816.
- Algorithm 1 & 2, $s_i$ and $p_i$ are vectors with length $H$, which should be clearly mentioned.
- Line 845, “in in”
- Line 852, “multi-objective optimization problems (MOPs)” -> “Multi-objective Optimization Problems (MOPs)”
- Line 856, “These solutions” -> “The objective vectors of these solutions”
- Line 859, “Pareto-optimal set (PS)” -> “Pareto-optimal Set (PS)”
- Line 875, add “if it is better than the current solution”
- Line 883, “is calculated $\bm{x}^{(i)}$” -> “$\bm{x}^{(i)}$ is calculated”
- Line 897, “the Algorithm 3” -> “Algorithm 3”
- Table 3, delete “in MOEA/D”?
- Line 959, delete “and produces”
- Table 4, caption, “IGD values of the compared methods with the correct order and reverse order (denoted as -r)”
- “Compute resource” -> “Computation resource”

**Quality:**

3

**Strengths And Weaknesses:**

Strengths
1. The studied problem of dynamic algorithm configuration, where multiple hyperparameters have a sequential order that widely exists in practice, which is challenging but has been explored very little. Thus, the topic of this paper is significant.
2. The paper proposes an effective method: Contextual sequential MMDP formulation and sequential advantage decomposition, where the action-order information can be utilized.
3. The theoretical analysis of the IGM property of the proposed sequential advantage decomposition is interesting. The proof is accomplished by a standard inductive hypothesis. I have checked the proofs and believe they are correct.
4. The experiments are generally good. The authors compared their proposed method with a wide range of methods, including existing sequential RL and advanced multi-agent RL methods. The empirical results on the artificial benchmark Seq-Sigmoid and its variants show the usefulness of considering the sequential order of adjusting hyperparameters and the robustness of the proposed method against the failure of adjusting some hyper-parameters. The results on tuning the four hyperparameters of the popular algorithm MOEA/D show the value of the proposed method in practice.

Weaknesses
1. Some experiment settings (e.g., the method of calculating the sparsity) and mathematical symbols (e.g, $\mathcal{N}(T/2,T/4,H)$) need to be further clarified.
2. Some experimental results need to be further analyzed, e.g., the rank using the reverse order is better than that using the correct order for SAQL and MAPPO with 9 objectives in Table 4.
3. The generalization ability of MOEA/D can be enhanced by training and testing on different numbers of objectives.
4. No guidance to determine the correct adjustment order of the hyperparameters of an algorithm.

---

> ### Author Rebuttal · Authors · 2025-07-31
>
> Thanks for your valuable and encouraging comments! Below please find our response.
>
> ## Q1 Why can SADN avoid the long sequence issue
>
> Thanks for your comments. We apologize for not making this clear. The advantage decomposition in our SADN allows each individual advantage net to be updated independently and simultaneously, reducing their mutual influence and compounding errors. In contrast, ACE sequentially updates the Q nets based on the output of the previous agent (i.e., more centralized), which leads to compounding errors along the update chain, especially worsening in the long sequence updates. This phenomenon is also observed as "error amplification" [1-2], which often happens due to inter-agent interference [3]. One way to alleviate this phenomenon in the MARL community is to decompose the team value function into agent-wise value functions so that the individual agent can learn relatively independently [4]. As demonstrated in [4], the centralized approach consistently fails on relatively simple cooperative problems in practice due to the mutual influence of policy learning and the lack of reward allocation. Furthermore, [5] shows that the more centralized methods tend to suffer from scalability. To address this issue, our SADN decomposes the global advantage function sequentially, where the individual advantages are learned relatively independently, and the action sequence information is also captured. Experimental results in Figures 2-3 show that SADN significantly outperforms ACE. We will revise the paper to provide more analyses and clarify this point. Thank you very much!
>
> [1] Is Behavior Cloning All You Need? Understanding Horizon in Imitation Learning. NeurIPS, 2024.
>
> [2] An Outlook on the Opportunities and Challenges of Multi-Agent AI Systems. arXiv, 2025.
>
> [3] Model-Based Diagnosis of Multi-Agent Systems: A Survey. AAAI, 2022.
>
> [4] Value-Decomposition Networks for Cooperative Multi-Agent Learning. AAMAS, 2018.
>
> [5] QTRAN: Learning to Factorize with Transformation for Cooperative Multi-Agent Reinforcement learning. ICML, 2019.
>
>
> ## Q2 How to determine the right order of hyperparameters in practice?
>
> Thank you for the insightful and constructive question! In practice, there is an intuitive and natural way to determine the correct order: setting it according to the sequence in which the hyperparameters take effect during the execution of the algorithm. Specifically, in the code implementation of the target algorithm, all hyperparameters take effect in a specific order, which is the order we have set. For example, in MOEA/D, after initializing the weights, we first determine the neighborhood subproblems (agent 1: neighborhood size), then optimize the subproblems using certain operators (agent 2: types of reproduction operators and agent 3: parameters of reproduction operators), and finally decide whether to adjust the weights (agent 4: weights) based on the results of this generation.
>
> This strategy aligns with our intuition and performs well in the experiments. However, we acknowledge that although it can be a good choice, this order may not always be the optimal one for training efficiency and effectiveness, as we stated in the conclusion. Determining the best order for the dynamic configuration of hyperparameters in general algorithms can be challenging, and we leave it as our future work. This may involve applying large language models and leveraging their knowledge to set the proper order, as well as using causal models to automatically learn the hyperparameters' causal structure from the data. We will revise to make this part clear. Thank you.
>
> ## Q3 Why can the symmetrical formulation in the Seq-Sigmoid avoid the influence on the former hyperparameter?
>
> Thank you for your valuable comment. In our proposed Seq-Sigmoid, the previous agent determines the scaling factor of the next agent by choosing an action greater or less than 0.5. To eliminate the impact caused by different choices of scaling factors, we design the reward value to be symmetric around 0.5. This symmetry ensures that, regardless of the scaling factor selected, there is always an optimal reward value for the current agent. As a result, the agent does not need to sacrifice its own optimal reward to accommodate the optimization of subsequent agents—for instance, it no longer has to choose a suboptimal value in order to adjust the scaling factor. To clarify this further, we will add specific examples and reward landscape figures in the revised paper. Thank you again!
>
> ## Q4 Explanation for $\mathcal{N}(T/2, T/4, H)$ and the suggestion to clearly mention $n<H$.
> Thank you for the question and suggestion. $\mathcal{N}(T/2, T/4, H)$ means we have $H$ parameters which all follow a Gaussian distribution with a mean of $T/2$ and a variance of $T/4$. In our setting, we always have $n<H$. We will revise to make it clear. Thank you.
>
> ## Q5 Concerns about the sufficiency of the three runs in Figure 2 and 3.
>
> Thank you for the valuable question. We believe that three runs are sufficient to support our conclusion for the following reasons:
>
> - In our experimental setup, we resample the instance in each episode, meaning that every point on the training curve reflects the generalization ability of the learned policy at that specific training phase. We believe this setting significantly reduces the impact of randomness across different runs.
>
> - As shown in Figures 2 and 3, the variance of our method is very small, as is that of most well-performing methods, and our method clearly exhibits better performance and robustness on more challenging tasks.
>
> To further address your concern, we have conducted additional runs. However, we could not include the detailed results due to space limitation. The relative performance of the algorithms and the experimental conclusions remain unchanged. We will include these results into our revised paper. Thank you.
>
>
> ## Q6 Further examining the generalization ability.
>
> Thank you for the insightful question! Yes, we maintain the dimensions consistent in one series of experiments' training and testing. We agree with you that testing with different dimensions can further examine the generalization ability. The results are shown below:
>
> | | MOEA/D | SADN-6 | SADN-9 |SADN-12|SADN|
> |---|---|---|---|---|---|
> |average rank|3.708|2.083|2.5|1.708|1.542|
>
> Due to space limitations, we only report the average rank of several key methods. SADN-n refers to the model trained exclusively on problems with n dimensions. Overall, the results demonstrate that our proposed method exhibits good generalization ability across different dimensions. However, we acknowledge that cross-dimensional generalization remains a significant challenge, and we will further investigate and improve its generalization capability in the future, such as through multi-dimensional mixed training and employing multi-head models. We will incorporate these discussions into our paper. Thank you very much!
>
>
> ## Q7 Introduce the IGD metric and the parameter order of MOEA/D in the main paper; other revision suggestions.
>
> Thank you for the valuable suggestions. We will revise the paper carefully according to your suggestions to include these contents.
>
> ## Q8 How to calculate the sparsity of each solution when the size of the elite population exceeds the capacity?
>
> Thank you for the question. As we mentioned in line 892 and 893, every solution in the elite population has its sparsity level calculated with respect to the current population, and when a new solution is about to be added to the elite population, the solution with the lowest sparsity level in the elite population will be removed immediately and the size of the elite population remains the same. Therefore, we will not calculate the sparsity of each solution when the size of the elite population exceeds the capacity.
>
> ## Q9 Concerns about fair comparisons between VDN, QMIX, and MAPPO with respect to the default hyperparameter settings.
>
> Thank you for your question. For a fair comparison, we try our best to maintain hyperparameter consistency among the algorithms of the same type. For example, for the value-based methods like SADN, ACE, SAQL, VDN, and QMIX, we use the same hidden layer size, learning rate, batch size, etc. The detailed settings can be found in Table 3 of Appendix B.3. Since different types of algorithms have different components (e.g., value-based methods like VDN vs. policy gradient-based methods like MAPPO), we follow the suggested hyperparameter settings in EPyMARL, which is one of the most popular open-source frameworks for MARL. Therefore, we believe that our hyperparameter settings are fair.
>
>
> ## Q10 Explain why for SAQL and MAPPO, the rank using the reverse order is better than that using the correct order?
>
> Thank you for your insightful question. This is because these two methods cannot efficiently make full use of the action sequence information, causing the performance of the correct order to occasionally be far from optimal, and sometimes even worse than the reverse order.
>
> - For MAPPO, it is not originally designed to capture the sequence information, and we basically add the previous agent's action to the next agent's state, according to [4].
>
> - For SAQL, as an extension to IQL [5], it lacks reward allocation and fails to reveal the interactions among the agents, which leads to a weak ability to make use of the action sequence information.
>
> [4] Revisiting some common practices in cooperative multi-agent reinforcement learning. ICML, 2022.
>
> [5] Multiagent cooperation and competition with deep reinforcement learning. PLoS ONE.
>
> ## Q11 Paper formatting concerns and typos
>
> Thank you very much for pointing these out! We will carefully revise them according to your comments.
>
> ---
>
> **We hope that our response has addressed your concerns, but if we missed anything, please let us know.**

---

> > ### Comment · Reviewer_HrJN · 2025-08-01
> >
> > Thanks for the detailed response. My questions have been addressed well. I am happy to increase my score by 1.

---

> > > ### Author Response · Authors · 2025-08-05
> > >
> > > Thanks for your feedback! We are glad to hear that your concerns have been addressed. We will ensure that the additional results are included in the final version, and we will carefully revise our paper according to your valuable suggestions. Thank you.

---

### Decision · Program_Chairs · 2025-09-17

**Decision:**

Accept (poster)

**Comment:**

Three reviewers recommended accepting the paper, with one weakly accepting it. The reviewers found the topic of the paper to be relevant, the method to be effective and the experiments to be convincing. I recommend to accept the paper.